# Variational regional inverse modeling of reactive species emissions with PYVAR-CHIMERE-v2019

Audrey Fortems-Cheiney[1], Isabelle Pison[1], Grégoire Broquet[1], Gaëlle Dufour[2], Antoine Berchet[1], Elise Potier[1], Adriana Coman[2], Guillaume Siour[2], and Lorenzo Costantino[2]

[1]Laboratoire des Sciences du Climat et de l'Environnement, LSCE-IPSL (CEA-CNRS-UVSQ), Université Paris-Saclay, 91191 Gif-sur-Yvette, France.

[2]Laboratoire Interuniversitaire des Systèmes Atmosphériques, UMR CNRS 7583, Université Paris Est Créteil et Université Paris Diderot, Institut Pierre Simon Laplace, Créteil, France.

## Abstract

Up-to-date and accurate emission inventories for air pollutants are essential for understanding their role in the formation of tropospheric ozone and particulate matter at various temporal scales, for anticipating pollution peaks and for identifying the key drivers that could help mitigate their concentrations. This paper describes the Bayesian variational inverse system PYVAR-CHIMERE, which is now adapted to the inversion of reactive species. Complementarily with bottom-up inventories, this system aims at updating and improving the knowledge on the high spatio-temporal variability of emissions of air pollutants and their precursors. The system is designed to use any type of observations, such as satellite observations or surface station measurements. The potential of PYVAR-CHIMERE is illustrated with inversions of both CO and $NO_x$ emissions in Europe, using the MOPITT and OMI satellite observations, respectively. In these cases, local increments on CO emissions can reach more than +50%, with increases located mainly over Central and Eastern Europe, except in the south of Poland, and decreases located over Spain and Portugal. The illustrative cases for $NO_x$ emissions also lead to large local increments (> 50%), for example over industrial areas (e.g., over the Po Valley) and over the Netherlands. The good behavior of the inversion is shown through statistics on the concentrations: the mean bias, RMSE, standard deviation and correlation between the simulated and observed concentrations. For CO, the mean bias is reduced by about 27% when using the posterior emissions, the RMSE and the standard deviation are reduced by about 50% and the correlation is strongly improved (0.74 when using the posterior emissions against 0.02); for $NO_x$, the mean bias is reduced by about 24%, the RMSE and the standard deviation are reduced by about 7% but the correlation is not improved. We reported strong non-linear relationships between $NO_x$ emissions and satellite $NO_2$ columns, now requiring a fully comprehensive scientific study.

## 1. Introduction

The degradation of air quality is a worldwide environmental problem: 91% of the world's population have breathed polluted air in 2016 according to the World Health Organization (WHO), resulting in 4.2 millions of premature deaths every year [WHO, 2016]. The recent study of Lelieveld et al. [2019] even suggests that the health impacts attributable to outdoor air pollution are substantially higher than previously assumed (with 790,000 premature deaths in the 28 countries of the European Union against the previously estimated 500,000 [EEA, 2018]). The main regulated primary (i.e. directly emitted in the atmosphere) anthropogenic air pollutants are carbon monoxide (CO), nitrogen oxides ($NO_x$ =$NO+NO_2$), sulfur dioxide ($SO_2$), ammonia ($NH_3$), volatile organic compounds (VOCs), and primary particles. These primary air pollutants are precursors of secondary (i.e. produced in the atmosphere through chemical reactions) pollutants such as ozone ($O_3$) and Particulate Matter (PM), which are also threatening to both human health and ecosystems. Monitoring concentrations and quantifying emissions are still challenging and limit our capability to forecast air quality to warn population and to assess i) the exposure of population to air pollution and ii) the efficiency of mitigation policies.

Bottom-up (BU) inventories are built in the framework of air quality policies such as The Convention on Long-Range Transboundary Air Pollution (LRTAP, http://www.unece.org) for air pollutants. Based on national annual inventories, research institutes compile gridded global or regional, monthly inventories (mainly for the US, Europe and China) with a high spatial resolution (currently regional or city scale inventories are typically finer than 0.1°x0.1°). These inventories are constructed by combining available (economic) statistics data from different detailed activity sectors with the most appropriate emission factors (defined as the average emission rate of a given species for a given source or process, relative to the unit of activity in a given administrative area). It is important to note that the activity data (often statistical data) has an inherent uncertainty and that its reliability may vary between countries or regions. In addition, the emission factors bear large uncertainties in their quantification [Kuenen et al., 2014; EMEP/EEA, 2016; Kurokawa et al., 2013]. Moreover, these inventories are often provided at the annual or monthly scale with typical temporal profiles to build the weekly, daily and hourly variability of the emissions. The combination of uncertain activity data, emission factors and emission timing can be a large source of uncertainties, if not errors, for forecasting or analyzing air quality [Menut et al., 2012]. Finally, since updating the inventories and gathering the required data for a given year is costly in time, manpower and money, only a few institutes have offered estimates of the gaseous pollutants for each year since 2011 (i.e, European Monitoring and Evaluation Programme EMEP updated until the year 2017, MEIC updated until the year 2017 to our knowledge). Nevertheless, using knowledge

from inventories and air quality modeling, emissions have been mitigated. For example, from 2010 to nowadays, emissions in various countries have been modified and/or regional trends have been reversed downwards (e.g., the decrease of $NO_x$ emissions over China since 2011 [de Foy et al., 2016]), leading to significant changes in the atmospheric composition. Consequently, the knowledge of precise and updated budgets, together with seasonal, monthly, weekly and daily variations of gaseous pollutants driven, amongst other processes, by the emissions are essential for understanding their role in the formation of tropospheric ozone and PMs at various temporal scales, for anticipating pollution peaks and for identifying the key drivers that could help mitigate these concentrations.

In this context, complementary methods have been developed for estimating emissions using atmospheric observations. They operate in synergy between a chemistry-transport model (CTM) which links the emissions to the atmospheric concentrations, atmospheric observations of the species of interest, and statistical inversion techniques. A number of studies using inverse modeling were first carried out for long-lived species such as greenhouses gases (GHGs) (e.g., carbon dioxide $CO_2$ or methane $CH_4$) at the global or continental scales [Hein et al., 1997; Bousquet et al. 1999], using surface measurements. Later, following the development of monitoring station networks, the progress of computing power, and the use of inversion techniques more appropriate to non-linear problems, these methods were applied to shorter-lived molecules such as CO. For these various applications (e.g., for $CO_2$, $CH_4$, CO), the quantification of sources was solved at the resolution of large regions [Pétron et al., 2002]. Finally, the growing availability and reliability of observations since the early 2000s (in-situ surface data, remote sensing data such as satellite data), the improvement of the global CTMs, of the computational capacities and of the inversion techniques have increased the achievable resolution of global inversions, up to the global transport model grid cells, i.e. typically with a spatial resolution of several hundreds of square kilometers [Stavrakou and Muller, 2006; Pison et al., 2009; Fortems-Cheiney et al., 2011; Hooghiemstra et al., 2012; Yin et al., 2015; Miyazaki et al., 2017, Zheng et al., 2019].

Today, the scientific and societal issues require an up-to-date quantification of pollutant emissions at a higher spatial resolution than the global one and imply to widely use regional inverse systems. However, although they are suited to reactive species such as CO and $NO_x$, and their very large spatial and temporal variability, they have hardly been used to quantify pollutant emissions. Some studies inferred $NO_x$ [Pison et al., 2007; Tang et al., 2013] and VOC emissions [Koohkan et al., 2013] from surface measurements. Konovalov et al. [2006, 2008, 2010], Mijling et al. [2012, 2013], van der A et al. [2008], Lin et al. [2012] and Ding et al. [2017] have also shown that satellite

observations are a suitable source of information to constrain $NO_x$ emissions. These regional
inversions using satellite observations were often based on Kalman Filter (KF) schemes [Mijling et
al., 2012, 2013; van der A et al., 2008; Lin et al., 2012; Ding et al., 2017].

Variational inversion systems allow solving for high dimensional problems, typically solving for
the fluxes at high spatial and temporal resolution, which can be critical to fully exploit satellite
images. Here, we present the Bayesian variational atmospheric inversion system PYVAR-
CHIMERE for the monitoring of anthropogenic emissions of reactive species at the regional scale.
It is based on the Bayesian variational assimilation code PYVAR [Chevallier et al. 2005] and on the
regional state-of-the-art CTM CHIMERE [Menut et al., 2013; Mailler et al., 2017]. CHIMERE is
dedicated to the study of regional atmospheric pollution events [e.g., Ciarelli et al., 2019; Menut et
al., 2020], included in the operational ensemble of the Copernicus Atmosphere Monitoring Service
(CAMS) regional services. The main strengths of PYVAR-CHIMERE come from the strengths of
CHIMERE and from its high modularity for the definition of the control vector. CHIMERE is
indeed an extremely flexible code, in particular for the definition of the chemical scheme.
The PYVAR-CHIMERE system takes advantage of the previous developments for the
quantification of fluxes of long-lived GHG species such as $CO_2$ [Broquet et al., 2011] and $CH_4$
[Pison et al., 2018] at the regional to the local scales, but now solves for reactive species such as CO
and $NO_x$. It has also a better level of robustness, clarity, portability, and modularity than these
previous systems. Variational techniques require the adjoint of the model to compute the sensitivity
of simulated atmospheric concentrations to corrections of the fluxes. CHIMERE is one of the few
CTMs for which the adjoint has been coded. For global models, they include: GEOS-CHEM
[Henze et al., 2007], IMAGES [Stavrakou and Muller, 2006], TM5 [Krol et al., 2008], GELKA
[Belikov et al., 2016] and LMDz [Chevallier et al., 2005; Pison et al., 2009] ; for limited-area
models they include: CMAQ [Hakami et al., 2007], EURAD-IM [Elbern et al., 2007],
RAMS/CTM-4DVAR [Yumimoto et Uno, 2006], WRF-CO2 4D-Var [Zheng et al., 2018]).

The principle of variational atmospheric inversion and the configuration of PYVAR-CHIMERE are
described in Section 2 and in Section 3, respectively. Details about the forward, tangent-linear and
adjoint codes of CHIMERE are also given. Then, the potential of PYVAR-CHIMERE is illustrated
in Section 4 with the optimization of European CO and $NO_x$ emissions, constrained by observations
from the Measurement of Pollution in the Troposphere (MOPITT) and from the Ozone Monitoring
Instrument (OMI) satellite instruments, respectively.


## 2. Principle of Bayesian variational atmospheric inversion

In what follows, we use the notations and equations used in the inverse modeling community [Rayner et al., 2019]. The Bayesian variational atmospheric inversion method adjusts a set of control parameters, including parameters related to the emissions whose estimate is the primary target of the inversion.

The prior information about the parameters $\mathbf{x}$ to be optimized during the inversion process is given by the vector $\mathbf{x^b}$. The parameters to be optimized can be surface fluxes but may also include initial or boundary conditions for example, as explained in Section 3.4. The adjustments are applied to prior values, usually taken, for the emissions, from pre-existing BU inventories. The principle is to minimize, on the one hand, the departures from the prior estimates of the control parameters, which are weighted by the uncertainties in these estimates (called hereafter "prior uncertainties"), and, on the other hand, the differences between simulated and observed concentrations, which are weighted by all other sources of uncertainties explaining these differences (called hereafter all together "observation errors"). In statistical terms, the inversion searches for the most probable estimate of the control parameters given their prior estimates, observations, CTM and their associated uncertainties. The solution, which will be called posterior estimate, is found by the iterative minimization of a cost function $J$ [Talagrand et al., 1997], defined as:

$$J(\boldsymbol{x}) = \left(\boldsymbol{x} - \boldsymbol{x^b}\right)^T \boldsymbol{B}^{-1}\left(\boldsymbol{x} - \boldsymbol{x^b}\right) + (H(\boldsymbol{x}) - \boldsymbol{y})^T \boldsymbol{R}^{-1}(H(\boldsymbol{x}) - \boldsymbol{y}) \qquad \text{(Eq. 1)}$$

$H$ is the non-linear observation operator that projects the control vector $\mathbf{x}$ onto the observation space. In most of the variational atmospheric inversion cases (such as those described in Section 4), the observation operator includes the operations performed by the CTM in linking the emissions to the concentrations and any other transformation to compute the simulated equivalent of the observations such as an interpolation or an extraction and averaging of the simulated concentration fields (see Section 3.5). The observations in $\mathbf{y}$ could be surface measurements and/or remote sensing data such as satellite data. The prior uncertainties and the observation errors are assumed to be unbiased and to have a Gaussian distribution. Consequently, the prior uncertainties are characterized by their covariance matrix $\mathbf{B}$ and the observation errors are characterized by their covariance matrix $\mathbf{R}$. By definition, the observation errors combine errors in both the data and the observation operator, in particular  measurement errors and errors in the conversion of satellite measurement into concentration data, errors from the CTM, representativity errors due to the comparison between point measurements and gridded models or due to the representation of the fluxes as gridded maps at a given spatial resolution, and aggregation errors associated with the optimization of emissions at a given spatial and/or temporal resolution (as specified in the control vector) that is different from (usually coarser than) that of the CTM [Wang et al., 2017].


For inversions with observation and control vectors having a high dimension, the minimum of $J$
cannot be found analytically due to computational limitations. It can be reached iteratively with a
descent algorithm. In this case, the iterative minimization of $J$ is based on a gradient method. $J$ is
calculated with the forward observation operator (including the CTM) and its gradient relative to
the control parameters **x** is provided by the adjoint of the observation operator (including the adjoint
of the CTM). The gradient is defined as:
$\nabla J(\boldsymbol{x}) = \boldsymbol{B}^{-1}(\boldsymbol{x} - \boldsymbol{x^b}) + H^* \boldsymbol{R}^{-1}(H(\boldsymbol{x}) - \boldsymbol{y})$ (Eq. 2)
where $H^*$ is the adjoint of the observation operator.

The high non-linearity of the chemistry for reactive species makes it difficult to use its tangent-
linear to approximate the actual observation operator, and, more generally, it makes the inversion
problem highly non-linear. Therefore, in PYVAR-CHIMERE, we use the M1QN3 limited memory
quasi-Newton minimization algorithm [Gilbert and Lemaréchal, 1989], which relies on the actual
CHIMERE non-linear model to compute $J$ at each iteration of the minimization. As most quasi-
Newton methods, it requires an initial regularization of **x**, the vector to be optimized, for better
efficiency. We adopt the most generally used regularization, made by minimizing in the space
defined by:
$\chi = \boldsymbol{B}^{\frac{1}{2}}(\boldsymbol{x} - \boldsymbol{x^b})$ (Eq. 3)
instead of the control space defined by **x**. Although more advanced regularizations can be chosen,
the minimization with $\chi$ is preferred for its simplifying the equation to solve. In the $\chi$-space,
Equation 2 can be re-written as follows:
$\nabla J \chi = \chi + \boldsymbol{B}^{\frac{1}{2}} H^* (\boldsymbol{R}^{-1}(H(\boldsymbol{x}) - \boldsymbol{y}))$ (Eq. 4)

The criterion for stopping the algorithm is based on a threshold set on the ratio between the final
and initial gradient norms or on the maximum number of iterations to perform. As shown in Figure
1, the minimization algorithm repeats the forward-adjoint cycle to get an estimate close to the
optimal solution of the inversion problem for the control parameters. This approximation of the
optimal estimate is found by satisfying the convergence criteria of the minimizer with a given
reduction of the norm of the gradient of $J$. Nevertheless, due to the non-linearity of the problem, the
minimization may reach a local minimum only, instead of the global minimum.

Finally, the calculation of the uncertainty in the estimate of emissions from the inversion, known as
"posterior uncertainty", is challenging in a variational inverse system [Rayner et al., 2019]. Even
though the posterior uncertainty can be explicitly written in various analytical forms, it requires the
inversion of matrices that are too large to invert given the current computational resources in our
variational approach. As a trade-off between computing resources and comprehensiveness, the
analysis error may be evaluated by an approach based on a propagation of errors through sensitivity
tests (e.g., as in Fortems-Cheiney et al., [2012]). It can also be estimated through a Monte Carlo
Ensemble [Chevallier et al., 2007], implemented in PYVAR. Nevertheless, it should be noted that
the cost of the Monte Carlo experiments used to derive these posterior uncertainties is huge.

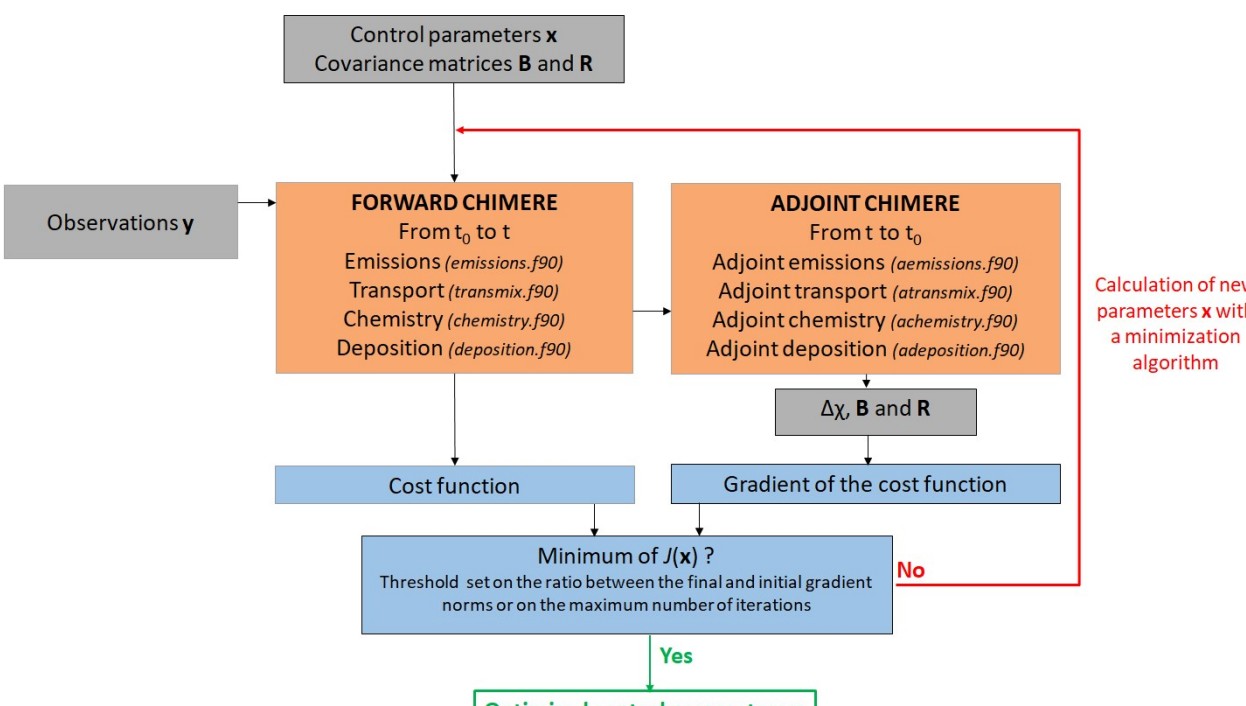

**Figure 1.** *Simplified scheme of the iterative minimization in PYVAR-CHIMERE. PYVAR,*
*CHIMERE and text sources are displayed in blue, in orange and in grey, respectively.*

## 3. The PYVAR-CHIMERE configuration

### 3.1. PYVAR adapted to CHIMERE

The PYVAR-CHIMERE inverse modeling system is based on the Bayesian variational assimilation
code PYVAR [Chevallier et al. 2005] and on a previous inversion system coupled to CHIMERE
[Pison et al., 2007]. PYVAR is an ensemble of Python scripts, which deals with preparing the
vectors and the matrices for the inversion, drives the required Fortran codes of the transport model
and computes the minimization of the cost function to solve the inversion. Previously used for
global inversions with the LMDz model [e.g., Pison et al., 2009; Chevallier et al., 2010; Fortems-
Cheiney et al., 2011; Yin et al., 2015; Locatelli et al., 2015; Zheng et al., 2019], PYVAR has been
adapted to CHIMERE with an adjoint code without chemistry by Broquet et al. [2011]. In order to
couple PYVAR to the new state-of-the-art version of CHIMERE (see Section 3.2), to include

chemistry, and to increase its modularity, flexibility and clarity, the new system described here has been developed. It includes elements of the inversion system (coded in Fortran90) of Pison et al. [2007].

### 3.2. Development and parallelization of the adjoint and tangent-linear codes of CHIMERE

To compute the sensitivity of simulated atmospheric concentrations to corrections to the fluxes, the adjoint of CHIMERE has been developed. Originally, the sequential adjoint was coded [Menut et al., 2000; Menut et al., 2003; Pison et al., 2007]. The adjoint has been coded by hand line by line, following the principles formulated by Talagrand [1997]. It contains exactly the same processes as the CHIMERE forward model. The code has been parallelized, which required a redesigning of the entire code, associated with a full testing scheme (see Section 3.3). Furthermore, the tangent-linear (TL) code has been developed and validated (see Section 3.3). Changes have been implemented in the forward CHIMERE code embedded in PYVAR-CHIMERE to match requirements of the studies conducted with this system. These changes have been implemented in both the adjoint and the TL codes. Compared to the CHIMERE 2013 version [Menut et al., 2013], the most important of these changes are, regarding geometry, the possibility of polar domains and the use of the coordinates of the corners of the cells instead of only the centers, allowing the use of irregular grids. Regarding transport, the non-uniform Van Leer transport scheme on the horizontal has been implemented, which is consistent with the use of irregular grids. Finally, various switches have been added to keep the system consistent for GHG studies. For example, we can avoid going into the chemistry, deposition or wet deposition routines when the focused species do not require them (e.g., no chemistry for methane or carbon dioxide at a regional scale).

PYVAR-CHIMERE is currently implemented with a full module of gaseous chemistry. As a compromise between the robustness of the method for reactive species, the time required coding the adjoint and the computational cost with a full chemical scheme, the aerosols modules of CHIMERE have not been included in the adjoint of CHIMERE yet and are therefore not available in PYVAR-CHIMERE. The development and maintenance of the adjoint means that the version used is necessarily one or two versions behind the distributed CHIMERE version (http://www.lmd.polytechnique.fr/chimere/). It should also be noted that PYVAR-CHIMERE only infers anthropogenic emissions at this stage. The optimization of biogenic emissions, which are linearly interpolated at the sub-hourly scale in CHIMERE, is currently under development.

As an example, Figure 2 presents a simplified scheme of how PYVAR scripts are used to drive this version of CHIMERE for forward simulations and inversions using satellite observations.

**Figure 2.** *Simplified scheme of how PYVAR scripts are used to drive CHIMERE for an inversion using satellite observations. PYVAR, CHIMERE and text sources are displayed in blue, in orange and in grey, respectively. "AK" refers to Averaging Kernels as detailed in Section 3.5.*

### 3.3. Accuracy of tangent-linear and adjoint codes

Different procedures have been implemented to test the accuracy of the TL and adjoint codes. To test the linearity of the TL, we compute a Taylor diagnostic. It consists in computing the TL at $\mathbf{x_0}$ for given increments $\Delta \mathbf{x}$, $dH\mathbf{x_0}(\Delta \mathbf{x})$, then the TL at $\mathbf{x_0}$ for $\lambda \times \Delta \mathbf{x}$ with $\lambda$ an arbitrary small number, $dH\mathbf{x_0}(\lambda \Delta \mathbf{x})$. Theoretically, if the TL is well coded, $\lambda dH\mathbf{x_0}(\Delta \mathbf{x}) = dH\mathbf{x_0}(\lambda \Delta \mathbf{x})$ by definition. In practice, the difference must be lower than 10 times the precision of the machine on which it is run.

The adjoint code is also tested, by verifying that $<H.\Delta \mathbf{x}, H.\Delta \mathbf{x}> = <\Delta \mathbf{x}, H^T H.\Delta \mathbf{x}>$ where $H^T$ stands for the adjoint at $\mathbf{x}$. What is actually computed is the ratio of the difference between the two scalar products to the second one and the accuracy of the computation. The difference should be a few times the precision of the machine on which it is run.

### 3.4. Definition of the control vector

The control vector is specified by the user in a text file. This file is formatted following Table 1. The parameters to be inverted may be fluxes and/or initial conditions and/or boundary concentration

conditions, at the grid-cell resolution or for one region encompassing up to the whole domain. Several types of corrections can be applied, they are defined in the code as "add", "mult" or "scale". Both the corrections "add" and "mult" are applied to gridded control variables. For correction type "add", the control variables are increments added to the corresponding components of the model inputs. For correction type "mult", the control variables are scaling factors multiplying the corresponding components of the model inputs. The difference between the two options "add" and "mult" plays a role when inverting fluxes which can switch from positive to negative values (like $CO_2$ natural fluxes). For type "scale", the control variables are scaling factors applied to maps different from the maps of emissions used as prior input of the forward model: for example, activity maps can be used and scaled to get emissions; the obtained values are then added to the corresponding components of the model inputs. With these various types, it is possible to define the control variables as the budgets of emissions for different regions, types of activities, and/or processes, which can thus be directly rescaled by the inversions, similarly to what is done in systems where the control vector is not gridded [Wang et al., 2018]).

Different simple but efficient ways of building the error covariance matrix **B** are implemented in PYVAR-CHIMERE. The variances and correlations are defined independently. The variances are specified by the user through the specification of the values for the corresponding standard deviation (i.e. the diagonal matrix of standard deviations $\sum$, Table 1) which can be made in terms of fixed values ("fx") or percentages ("pc"). For correction types "mult" and "scale", as well as for correction type "add" with a fixed value, the value is directly used as the uncertainty in the corresponding components of the control vector. For correction type "add" with a percentage provided, maps of standard deviation of uncertainty are built by applying this percentage to the matching input fields (fluxes, initial conditions, boundary conditions). The user may also provide a script to build personalized maps of variances.

Potential correlations between uncertainties in different types of control variables, e.g. between fluxes and boundary conditions, and correlations between uncertainties in different species, e.g. between fluxes of CO and $NO_x$, are not coded yet. Such correlations increase the observation constraint on the emissions in the inversion process by transferring information from one species to the other. The level (and sometimes the sign) and thus the impact on the inversion of such correlations highly depends on the study cases, and are often debated due to the lack of precise characterization of the uncertainties in inventories of anthropogenic emissions of GHG and pollutants [Super et al. 2020]. Only correlations for a given type of control variable and a given species are so far taken into account so that the **B** matrix is block diagonal. For a given type of

control variable and a given species (in the illustration in section 4.2.2: CO, NO or NO$_2$ fluxes),
spatial and temporal correlations can be defined using correlation lengths through time Lt and space
Ls. Those lengths are used to model temporal and/or spatial auto-correlations using an
exponentially decaying function: the correlation r between parameters and at a given location but
separated by duration $d(x_i, x_j)$, or at a given time but distant by $d(x_i, x_j)$ is given by $r(x_i, x_j) =$
$exp\left(\frac{-d(x_i, x_j)}{L}\right)(Eq.\,5)$ where $L = L_T \vee L_S$ is the corresponding correlation length. There is no
correlation between uncertainties in land and ocean flux. Note that the spatial correlations are
computed for each vertical level independently when dealing with control variables with vertical
resolution (3D fields of fluxes when accounting for emission injection heights, or boundary/initial
conditions).Vertical correlations in the uncertainties in such variables have not been coded yet.
Apart from this, the system assumes that temporal correlations and spatial correlations depend on
the time lag and distance but not on the specific time and location of the corresponding parameters.
It also assumes that the correlation between uncertainties at different locations and different time
can be derived from the product of the corresponding autocorrelation in time and space.
Each block of **B** can thus be decomposed based on Kronecker products: $\mathbf{B}=\sum C_t \otimes C_s \sum$ (Eq. 6) where
$\otimes$ is the Kronecker product, $C_t$ and $C_s$ are the temporal and spatial correlations, respectively. The
calculations involving $\mathbf{B}^{1/2}$ (in Eq. 3, Eq. 4) are simplified in PYVAR-CHIMERE using the Eigen-
decomposition of $C_t$ and $C_s$. Its square root can be calculated according to: $C_t^{1/2} = V_{Ct}D_{Ct}^{1/2}V_{Ct}^{T}$ (Eq.
7) (and similarly for $C_s$), where $V_{Ct}$ is the matrix with the Eigenvectors as columns, and $D_{Ct}$ is the
diagonal matrix of Eigenvalues of $C_t$. It is possible to chose a threshold under which the eigenvalues
are truncated when computing the spatial correlations in order to save computation time and
memory, but not when computing the temporal correlations.

| Constrained species | Correction type : - Add - Mult - Scale | Spatial resolution | Temporal resolution (in hours) | Input to constrain: -Fluxes -Initial conditions -Lateral Boundary conditions -Top Boundary conditions | B variance coefficient: -fx -pc | Decorrelation time (in hours) | Decorrelation length on land (in km) | Decorrelation length on sea (in km) |
|---|---|---|---|---|---|---|---|---|
| CO | add | 0.5°x0.5° | 168 | Fluxes | 100 % | - | - | - |
| CO | add | 0.5°x0.5° | 1 | Initial conditions | 15% | - | - | - |
| CO | add | 0.5°x0.5° | 168 | Lateral Boundary | 15% | - | - | - |

| | | | | conditions | | | | |
|---|---|---|---|---|---|---|---|---|
| CO | add | 0.5°x0.5° | 168 | Top Boundary conditions | 15% | - | - | - |
| NO | add | 0.5°x0.5° | 24 | Fluxes | 50 % | - | 50 | 50 |
| NO | add | 0.5°x0.5° | 1 | Initial conditions | 15% | - | - | - |
| NO$_2$ | add | 0.5°x0.5° | 24 | Fluxes | 50 % | - | 50 | 50 |
| NO$_2$ | add | 0.5°x0.5° | 24 | Initial conditions | 15% | - | - | - |

**Table 1**. *Examples for the definition of the control vector and for the construction of the B matrix, as illustrated in Section 4.*

### 3.5. Equivalents of the observations

During forward simulations, the equivalents of the components of **y** (i.e, the equivalents of the individual data) are calculated by PYVAR-CHIMERE. It includes the CTM and an interpolation (see below the vertical interpolation from the model's grid to the satellite levels) or an extraction and averaging (e.g. extracting the grid cell matching the geographical coordinates of a surface station and averaging over one hour). As a compromise between technical issues such as the time required for reading/writing files, the observation operator $H$ that generates the equivalent of the observations by the model (i.e. $H(\mathbf{x})$) has been so far partly embedded in the code of CHIMERE. It makes it easier to use finer time intervals than available in the usual hourly outputs of CHIMERE to compute the required information (e.g., within the finer CTM physical time steps).

To make comparisons between simulations and satellite observations, the simulated vertical profiles are first interpolated on the satellite's levels (with a vertical interpolation on pressure levels) in CHIMERE. Then, the averaging kernels (AKs), when available, are applied to represent the vertical sensitivity of the satellite retrieval. Two types of formula, depending on the satellite observations used, have been detailed in PYVAR-CHIMERE for the use of AKs: $\boldsymbol{c_m} = \boldsymbol{AK}.\boldsymbol{c_{m(o)}}$ (Eq. 8) or $\boldsymbol{c_m} = \boldsymbol{x_a} + \boldsymbol{AK}\left(\boldsymbol{c_{m(o)}} - \boldsymbol{x_a}\right)$ (Eq. 9) where $\mathbf{c_m}$ is the modeled column, AK contains the averaging kernels that can be provided in the form of vector (e.g., OMI product) or matrix (e.g., MOPITT product), $\boldsymbol{x_a}$ is the prior state vector (provided together with the AKs when relevant) and $\mathbf{c_{m(o)}}$ is the vertical distribution of the original model partial columns interpolated to the pressure grid of the AKs.

### 3.6. Numerical language

The PYVAR code is in Python 2.7, the CHIMERE CTM is coded in Fortran90. The CTM requires several numerical tools, compilers and libraries. The PYVAR-CHIMERE system was developed and tested using the software versions as described in Table 2.

|  |  | URL | Version |
|---|---|---|---|
| Software | Python | https://www.python.org/downloads/ | 2.7 |
|  | Fortran compiler ifort | https://software.intel.com/en-us/fortran-compilers | Composer-xe-2013.2.146 |
| Libraries or packages | UnidataNetCDF | https://www.unidata.ucar.edu/ | 3 |
|  | Open MPI | https://www.open-mpi.org/ | 1.10.5 |
|  | GRIB_API | https://confluence.ecmwf.int/display/GRIB/Releases | 1.14 |
|  | nco | http://nco.sourceforge.net/#Source | 4.6.3 |

**Table 2**. *URL addresses for the development and the use of the PYVAR-CHIMERE system and its modules.*

PYVAR-CHIMERE's computation time for one node of 10 CPUs is about 4h for 1 day of inversion (with ~10 iterations) for the European domain size of 101 (longitude) x 85 (latitude) x 17 (vertical levels) used in Section 4. As described in Menut et al. [2013] for CHIMERE, the model parallelization results from a Cartesian division of the main geographical domain into several sub-domains, each one being processed by a worker process. To configure the parallel sub-domains, the user has to specify two parameters in the model parameter file: the number of sub-domains for the zonal and meridian directions. The total number of CPUs used is therefore the product of these two numbers plus one for the master process. The optimal number of CPUs for the parallelization of the transport scheme depends on the size of the tiles and also of the technical characteristics of the machine, because of the time required to exchange halos.

### 4. Potential of PYVAR-CHIMERE for the inversion of CO and $NO_x$ emissions

The potential of the PYVAR-CHIMERE system to invert emissions of reactive species is illustrated with the inversion of CO and $NO_x$ anthropogenic emissions in Europe respectively based on MOPITT CO data and OMI $NO_2$ data. We have chosen to present an illustration of CO inversion over seven days, the first week of March 2015. Considering the short lifetime of $NO_x$ of a few hours [Valin et al., 2013; Liu et al., 2016], we have chosen to present illustration of $NO_x$ inversion over one day, 19 February 2015. These particular periods have been chosen as they present a representative number of super-observations during winter, and as the emissions are high during that period. All the information required by the system to invert CO and $NO_x$ emissions is listed in Table 1.

### 4.1. Data and model description

#### 4.1.1. Observations y

We use CO data from the MOPITT instrument [Deeter et al., 2019]. MOPITT has been flown onboard the NASA EOS-Terra satellite, on a low sun-synchronous orbit that crosses the equator at 10:30 and 22:30 LST. The spatial resolution of its observations is about 22x22 $km^2$ at nadir. It has been operated nearly continuously since March 2000. MOPITT CO products are available in three variants: thermal-infrared TIR only, near-infrared NIR only and the multispectral TIR-NIR product, all containing total columns and retrieved profiles (expressed on a ten-level grid from the surface to 100 hPa). We choose to constrain CO emissions with the MOPITT surface product for our illustration. Among the different MOPITTv8 products, we choose to work with the multispectral MOPITTv8-NIR-TIR one, as it provides the highest number of observations, with a good evaluation against in situ data from NOAA stations [Deeter et al., 2019]. The MOPITTv8-NIR-TIR surface concentrations are sub-sampled into "super-observations" in order to reduce the effect of errors that are correlated between neighboring observations: we selected the median of each subset of MOPITT data within each 0.5°×0.5° grid-cell and each physical time step (about 5-10 minutes). After this screening, 8437 "super-observations" remain in the 7-day inversion (from 10667 raw observations). It is important to note that the potential of MOPITT to provide information at a high temporal resolution, up to the daily scale, is hampered by the cloud coverage (see the blanks in Figure 5b).

The observational constraint on $NO_2$ emissions comes from the OMI QA4ECV tropospheric columns [Muller et al., 2016; Boersma et al., 2016, Boersma et al., 2017]. The Ozone Monitoring Instrument (OMI), a near-UV/Visible nadir solar backscatter spectrometer, was launched onboard EOS Aura in July 2004. It has been flown on a 705 km sun-synchronous orbit that crosses the Equator at 13:30 LT. Our data selection follows the criteria of the OMI QA4ECV data quality statement. As the spatial resolution of the OMI data is finer than that of the chosen CHIMERE model grid (13x24 $km^2$ against 0.5°×0.5°, respectively), the OMI tropospheric columns are sub-sampled into "super-observations" (median of the OMI data within the 0.5°×0.5° grid-cell and each physical time step and its corresponding AK).

#### 4.1.2 CHIMERE set-up

CHIMERE is run over a 0.5°×0.5° regular grid (about 50x50$km^2$) and 17 vertical layers, from the surface to 200hPa (about 12km), with 8 layers within the first two kilometers. The domain includes 101 (longitude) x 85 (latitude) grid-cells (15.5°W-35°E; 31.5°N-74°N, see Figure 3). CHIMERE is driven by the European Centre for Medium-Range Weather Forecasts (ECMWF) meteorological

forecast [Owens and Hewson, 2018]. The chemical scheme used in PYVAR-CHIMERE is MELCHIOR-2, with more than 100 reactions [Lattuati, 1997; CHIMERE 2017], including 24 for inorganic chemistry. The prior anthropogenic emissions for CO and $NO_x$ emissions are obtained from the TNO-GCHco-v1inventory [Super et al., 2020], the last update of the TNO-MACCII inventory [Kuenen et al., 2014]. This inventory is based on the EMEP/Centre on Emission Inventories and Projections (CEIP) official country reporting for air pollutants done in 2017. It is an inventory at $6x6km^2$ horizontal resolution. From the annual and national budgets, each sector is assigned to a specific proxy to quantify the spatial variability of the emissions within each country. Temporal profiles are also provided per Gridded Nomenclature For Reporting (GNFR) sector code (variations due to the month, weekday and hour). Following the Generation of European Emission Data for Episodes (GENEMIS) recommendations [Kurtenbach et al., 2001; Aumont et al., 2003], $NO_x$ emissions are speciated as 90% of NO, 9.2% of $NO_2$, and 0.8% of nitrous acid (HONO). The TNO-GHGco-v1 inventory has been aggregated to the CHIMERE grid.

The prior anthropogenic emissions for VOCs are obtained from the EMEP inventory [Vestreng et al., 2005; EMEP/CEIP website]. Biogenic emissions come from the Model of Emissions of Gases and Aerosols from nature (MEGAN) [Guenther et al., 2006]. Different climatological values from the LMDZ-INCA global model [Szopa et al., 2008] or from a Monitoring Atmospheric Composition and Climate (MACC) reanalysis are used to prescribe concentrations at the lateral and top boundaries and the initial atmospheric composition in the domain. Full access to and more information about the MACC reanalysis data can be obtained through the MACC-II web site (http://www.copernicus-atmosphere.eu). In order to ensure realistic fields of simulated CO and $NO_2$ concentrations from the beginning of the inversion period, runs have been preceded with a 10-day spin-up.

### 4.1.3. CO Sensitivity to emissions and to initial and boundary conditions

With its lifetime of about two months, CO could be strongly influenced by the initial and lateral boundary conditions prescribed in the CTM. In fact, as seen in Figure 4b, initial and boundary conditions provide a relatively flat background and the patterns which appear clearly over the background are linked to surface emissions (Figure 4a). To characterize the uncertainties in the concentration fields due to the initial and lateral boundary conditions, we performed a sensitivity test by using either climatological values from LMDZ-INCA or a MACC reanalysis: maximum relative differences in concentrations of about 15% over continental land are estimated (Figure 4c). The errors assigned to initial and boundary conditions in Section 4.2.2 are based on this sensitivity test.

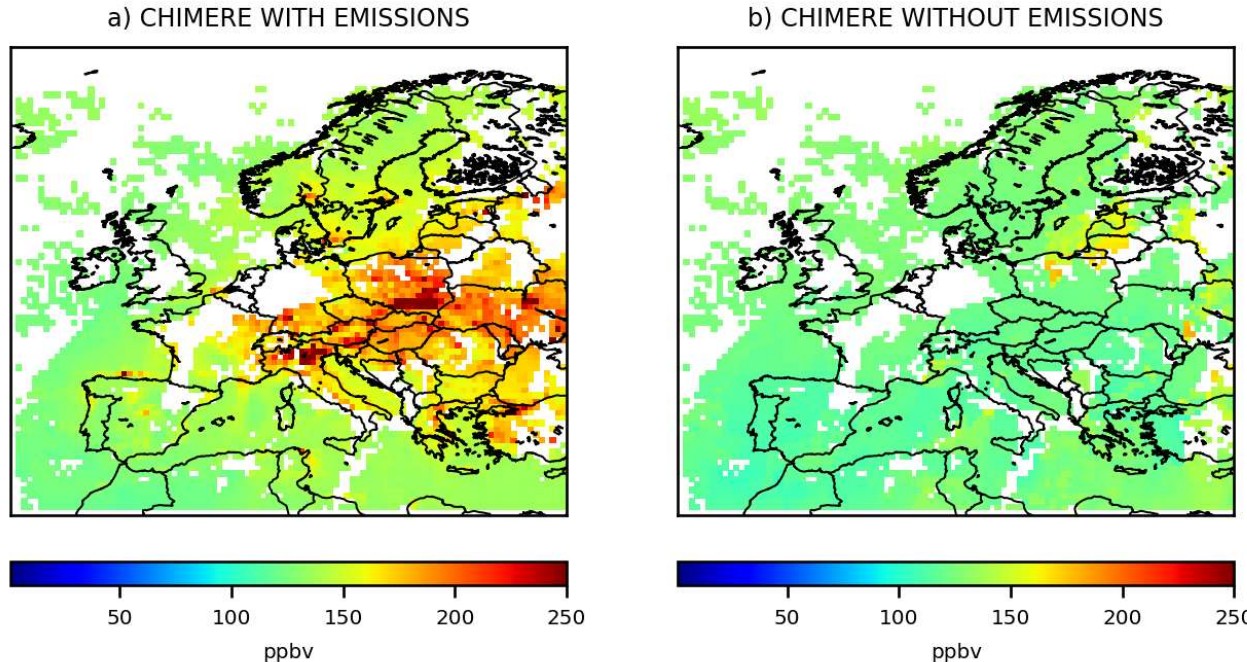


**Figure 3**. *Mean CO surface concentrations from the 1$^{st}$ to the 7$^{th}$, March 2015 simulated by CHIMERE a) with anthropogenic and biogenic emissions, and b) without emissions, in ppbv, at the 0.5°x0.5° grid-cell resolution.*

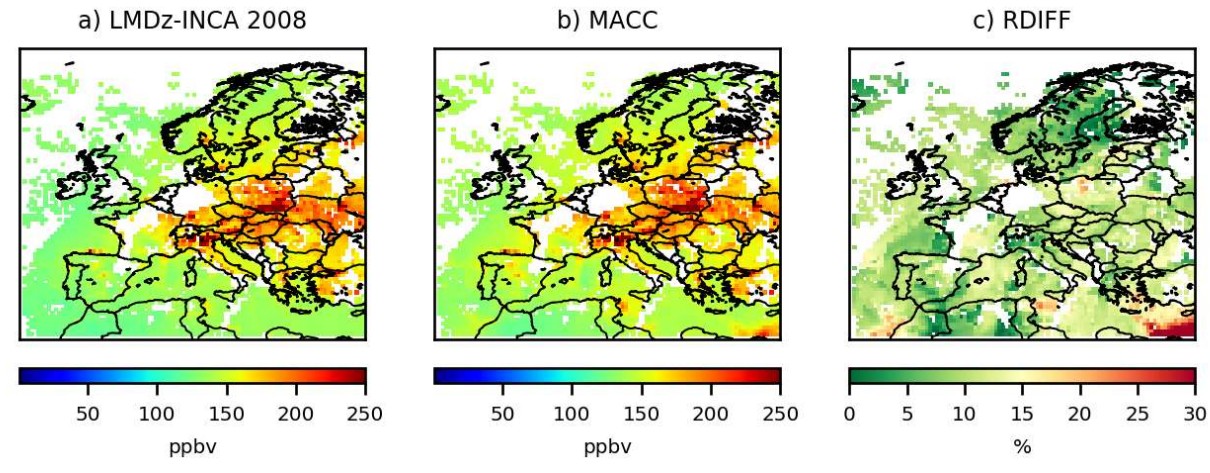


**Figure 4**. *Mean CO surface concentrations from the 1$^{st}$ to the 7$^{th}$, March 2015 simulated by CHIMERE using for initial and boundary conditions, a) the climatological values from the LMDZ-INCA global model  b) the climatological values from a MACC reanalysis, in ppbv, and c) the relative differences between these two simulations , in %, at the 0.5°x0.5° grid-cell resolution.*

### 4.1.4. Comparison between CHIMERE and the observations

Large discrepancies (Figure 5d) are found between the MOPITT CO observations (Figure 5b) and
the prior simulation by CHIMERE over Europe (Figure 5a). For the first week of March 2015, CO
concentrations are generally under-estimated by CHIMERE, particularly over Central and Eastern
Europe (excepted in the south of Poland). On the contrary, CO concentrations seem to be over-
estimated over Spain and Portugal. Large discrepancies are also found between the OMI $NO_2$ super-
observations and the prior simulation by PYVAR-CHIMERE (Figure 6d), as already noticed by
Huijnen et al. [2010], with an inter-comparison of $NO_2$ OMI-DOMINO tropospheric columns with
an ensemble of European regional air quality models including CHIMERE. Over Europe, the prior
simulation strongly underestimates the tropospheric columns over industrial areas (e.g., over the
Netherlands and over Po Valley). These discrepancies might be due to different causes, which can
all interact. A source of uncertainties is related to the observations. For example, satellite data inter-
comparison studies reveal large differences between different retrievals of the same compound [Qu
et al., 2020]. It can be explained by uncertainties from the CTM (e.g., through the underestimation
of the atmospheric production or the underestimation of the species lifetime). It could also be
explained by an underestimation of the anthropogenic emissions in the BU inventory.

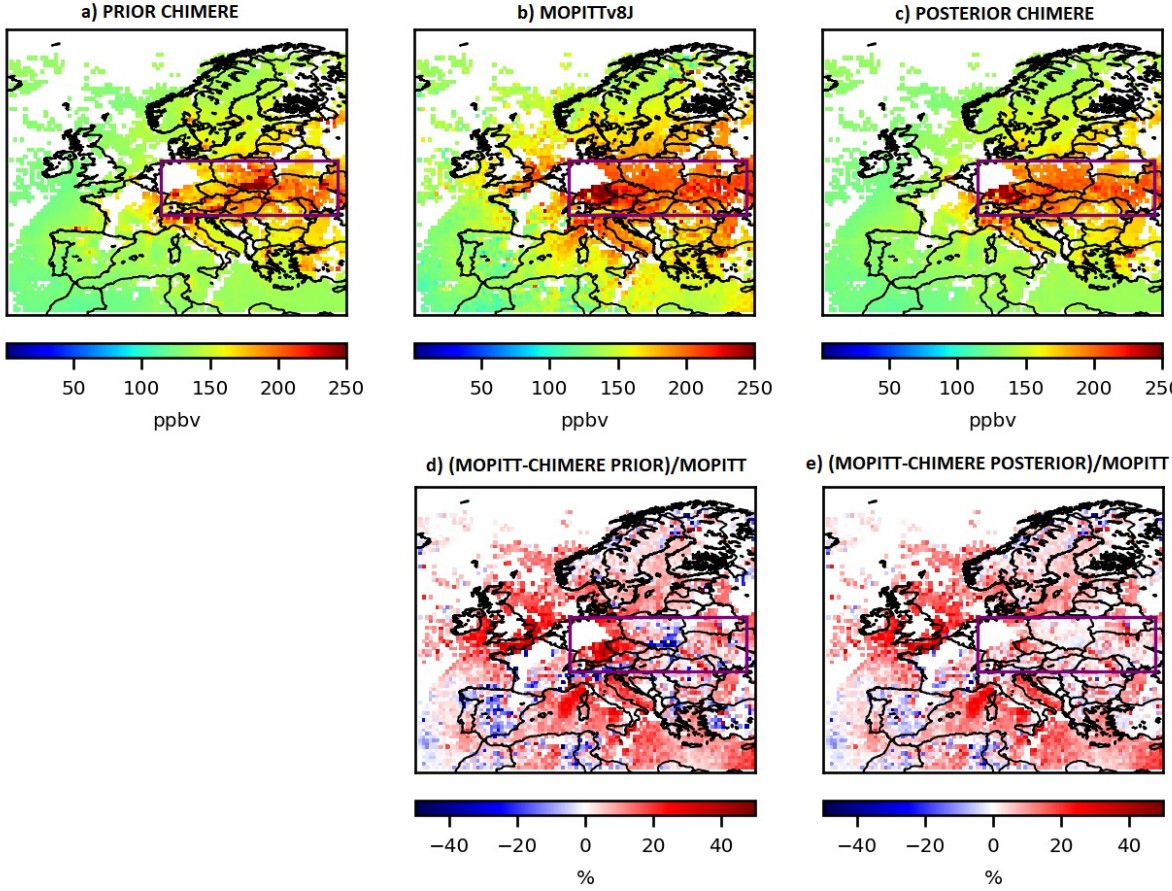

**Figure 5**. *Mean CO collocated surface concentrations from the 1$^{st}$ to the 7$^{th}$, March 2015 a) simulated by CHIMERE using the prior TNO-GHGco-v1 emissions and the climatological values from the LMDZ-INCA global model for initial and boundary conditions, b) observed by MOPITTv8-NIR-TIR and c) simulated by CHIMERE using the posterior emissions, in ppbv, at the*

*0.5°x0.5° grid-cell resolution. Relative differences between MOPITT and d) the prior CHIMERE simulation or e) the posterior CHIMERE simulation, in %. Statistics for the comparison between simulations and observations are given in Table 4 for the area in the purple box.*

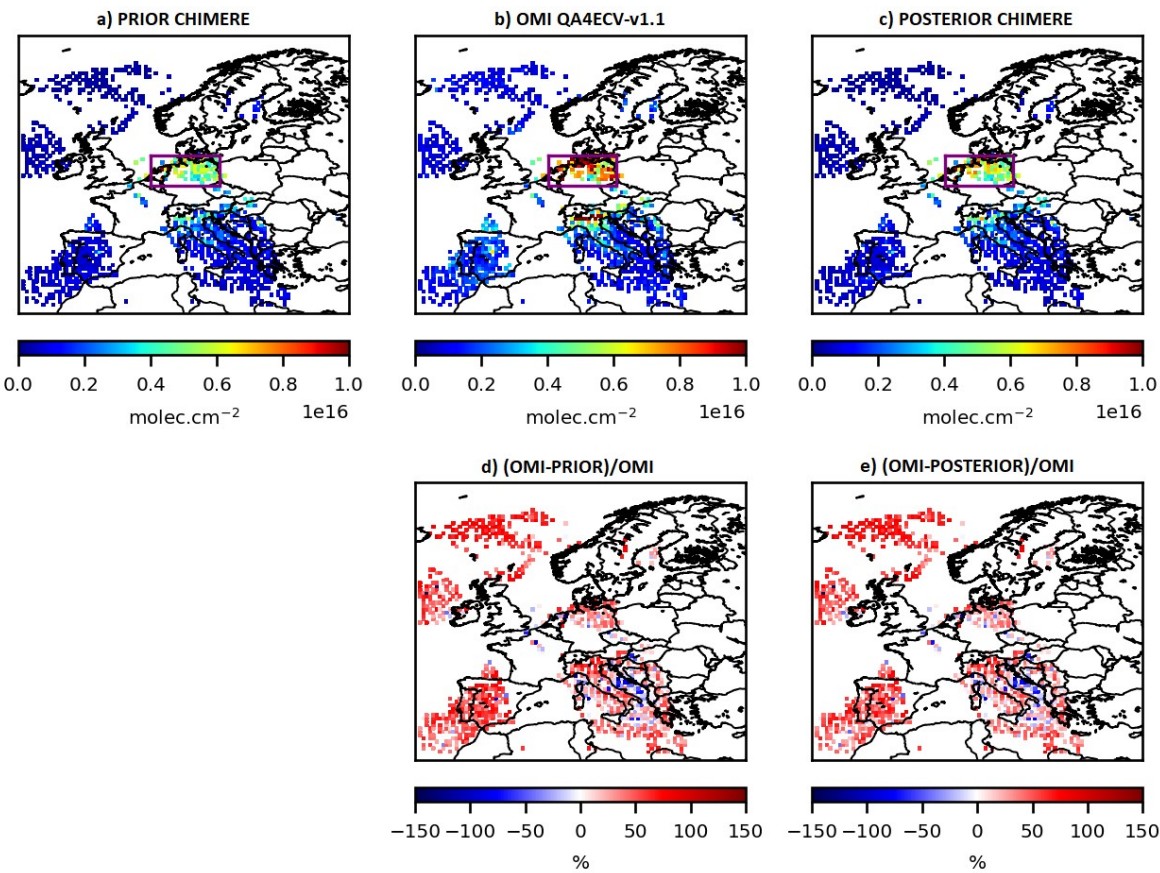

**Figure 6**. *NO$_2$ collocated tropospheric columns a) simulated by CHIMERE using the prior TNO-GHGco-v1 emissions and the climatological values from the LMDZ-INCA global model for initial and boundary conditions, b) observed by OMI and c) simulated by CHIMERE using the posterior emissions, in $10^{16}$ molec.cm$^{-2}$, at the 0.5°x0.5° grid-cell resolution, the 19$^{th}$, February 2015. Relative differences between OMI and d) the prior CHIMERE simulation or e) the posterior CHIMERE simulation, in %. Statistics for the comparison between simulations and observations are given in Table 5 for the area in the purple box.*

### 4.2. Inversions

#### 4.2.1. Control vector x

For the CO inversion, the control vector **x** is:

- the CO anthropogenic emissions at a 7-day temporal resolution, a 0.5° ×0.5° (longitude, latitude) horizontal resolution, and over the first 8 vertical levels, i.e. for each of the corresponding 101×85×8 grid cells,

- the CO lateral and top boundary conditions at a 7-day temporal resolution, at a 0.5° ×0.5° (longitude, latitude) resolution and over the 17 vertical levels of CHIMERE, i.e. (2x101 + 2x85) x 17 grid cells,
- the CO 3D initial conditions for the 1$^{st}$ March 2015 at 0:00 UTC, at a 0.5° ×0.5° (longitude, latitude) resolution, and over the 17 vertical levels of CHIMERE.

Considering its short lifetime, there is no boundary conditions for $NO_2$. For the $NO_x$ inversion, the control vector **x** is:

- the NO and $NO_2$ anthropogenic emissions at a 1-day temporal resolution, at a 0.5° ×0.5° (longitude, latitude) resolution and over the first 8 vertical levels, i.e. for each of the corresponding 101×85×8 grid cells,
- the NO and $NO_2$ 3D initial conditions for the 19$^{th}$ February 2015 at 0:00 UTC, at a 0.5° ×0.5° (longitude, latitude) resolution and over the 17 vertical levels of CHIMERE.

### 4.2.2. Covariance matrices B and R

To our knowledge, there are few available studies dealing with the estimates of the uncertainties in gridded bottom-up emission inventories at the 0.5°x0.5° resolution or higher. The characterization of their statistics in the inversion configuration is consequently often based on crude assumptions from the inverse modelers. Defining the covariance matrices **B** and **R** is not an easy task, while incorrectly specifying these matrices has a very strong impact on the results of the inversion. Especially, the relative weights of **B** and **R**, and the spatial and temporal correlations in B influence the degree of freedom and the structure for the adjustments attempted by the inversion in the optimization process. Consequently, as an example for the $NO_x$ inversion, different sensitivity tests described in Table 3 have been performed for the construction of the **B** matrix. For both the prior NO and $NO_2$ emissions at 1-day and 0.5° resolution, the prior error standard deviations are first assigned to 50% of the prior estimate of the emissions (test A), as in Souri et al. [2020]. Sensitivity tests have also been performed with prior error standard deviations assigned to 80 and 100% of the prior estimate of the emissions (test C and test D, respectively, Figure 8).

With prior error standard deviations set at 15% of the initial conditions, the changes in initial conditions are very small (not shown) and do not affect the posterior emissions (test B, Figure 8). As indicated in Section 3.4 and in Table 1, it is possible to use correlations in **B**, as in Broquet et al. [2011], in Broquet et al. [2013] and in Kadygrov et al. [2015]. We demonstrate the strong impact of spatial correlations, defined by an e-folding length of 50km over land and over the sea, on our inversions results (test E, Figure 8).

| Name of the sensitivity tests | Prior error standard deviations in B | | Spatial correlation in B | Number of iterations | Reduction of the norm of the gradient of $J$ |
|---|---|---|---|---|---|
| | On prior emissions | On prior initial conditions | | | |
| A | 50% | - | - | 4 | 99% |
| B | 50% | 15% | - | 6 | 98% |
| C | 80% | 15% | - | 7 | 97% |
| D | 100% | 15% | - | 6 | 95% |
| E | 50% | 15% | 50km | 5 | 92% |

**Table 3.** *Description of the different sensitivity tests performed for the construction of the **B** matrix*
*for the $NO_x$ inversion.*

Even though annual CO emissions in Western Europe may be well known, with uncertainties of 6%
according to Super et al., [2020], larger uncertainties could affect Eastern Europe. Moreover, large
uncertainties still affect bottom-up emission inventories at the 0.5° resolution: spatial
disaggregation of the national scale estimates to provide gridded estimates causes a significant
increase in the uncertainty for CO [Super et al., 2020]. For the inversion of CO emissions, the error
standard deviations assigned to the prior CO emissions at 7-day and 0.5° resolution are 100%. This
value of 100% has already been chosen in Fortems-Cheiney et al. [2011] and in Fortems-Cheiney et
al. [2012]. For this CO illustration, the covariance matrix **B** of the prior errors is defined as diagonal
(i.e. only variances in the individual control variables listed in 4.2.1 are taken into account).With
such a set-up, in theory, we could obtain negative posterior emissions since the inversion system
does not impose a constraint of positivity in the results. Nevertheless, even an uncertainty of 100%
leads to a prior distribution mostly (>80%) on the positive side. The assimilation of data showing an
increase above the background (at the edges of the domain; not shown) further drive the inversion
towards positive emissions for both CO and $NO_x$ inversions. In practice, our inversion does not lead
to negative posterior emissions (Figure 7b). Spatial and temporal correlations in **B** would further
limit the probability to get negative emissions locally by smoothing the posterior emissions at a
spatial scale at which the "aggregated" prior uncertainty is smaller than 100%. However, a
positivity constraint should be implemented in future versions of the system.

Based on the sensitivity test in Figure 4, the errors assigned to the CO lateral boundary conditions and to their initial conditions are set at 15%. As these relative errors are significantly lower than those for the emissions and as variations in the CO surface concentrations are mainly driven by emissions (Figure 3), we assume a small relative influence of the correction of initial and boundary conditions on our results. The variance of the individual observation errors in **R** is defined as the quadratic sum of the measurement error reported in the MOPITT and the OMI data sets, and of the CTM errors (including chemistry and transport errors and representativity errors) set at 20% of the retrieval values. The representativity errors could have been reduced with the choice of a finer CTM resolution (e.g., with a resolution closer to the size of the satellite pixel). Error correlations between the super-observations are neglected, so that the covariance matrix **R** of the observation errors is diagonal.

### 4.2.3. Inversion of CO emissions

Ten iterations are needed to reduce the norm of the gradient of *J* by 90% with the minimization algorithm M1QN3 and obtain the increments, i.e. the corrections provided by the inversion. The prior CO emissions over Europe for the first week of March 2015 and their increments are shown in Figure 7. As expected from the large differences (Figure 5d) between the prior surface concentrations (Figure 5a) and the MOPITT observations (Figure 5b), local increments can reach more than +50% (Figure 7b). CO emissions are increased over Central and Eastern Europe, except in the south of Poland. On the contrary, CO emissions are decreased over Spain and Portugal. The analyzed concentrations are the concentrations simulated by CHIMERE with the posterior fluxes: as expected, the optimization of the fluxes improves the fit of the simulated concentrations to the observations (Figure 5c and Figure 5e), particularly over Central and Eastern Europe. Over this area (see the purple box in Figure 5), the mean bias between the simulation and the observations has been reduced by about 27% when using the posterior emissions (mean bias of 11.6 ppbv against 15.9 ppbv with the prior emissions, Table 4). The RMSE and the standard deviation have been reduced by about 50% and the correlation has been strongly improved (0.74 when using the posterior emissions against 0.02).

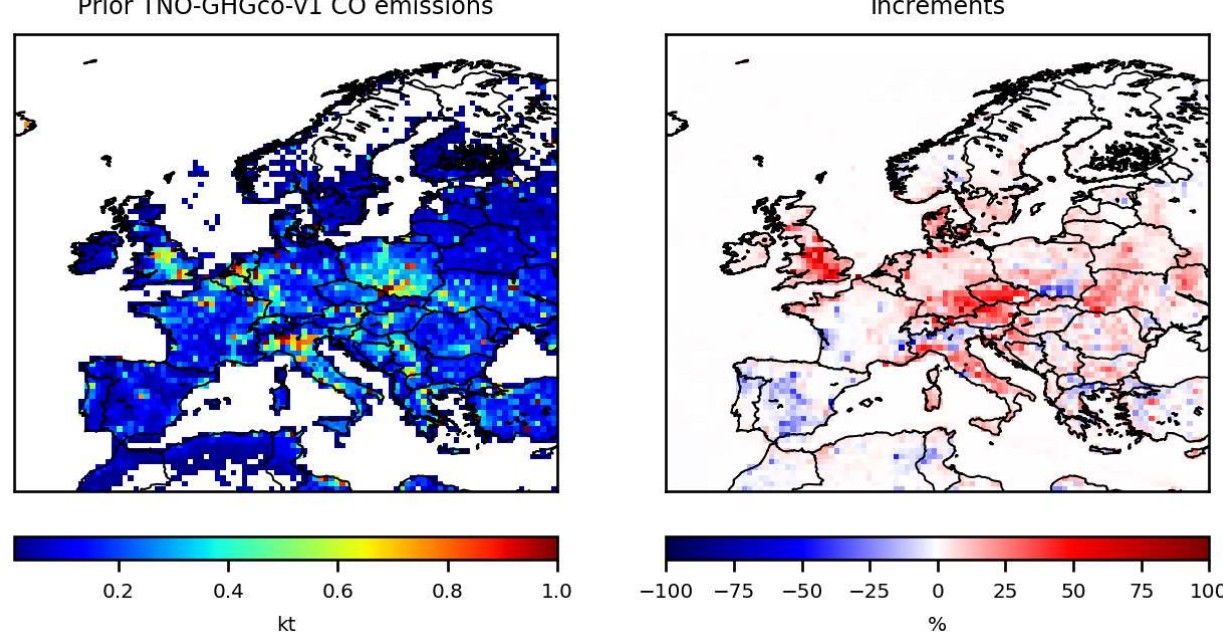

**Figure 7.** *a) TNO-GHGco-v1 CO anthropogenic prior emissions, in ktCO/grid-cell and b) increments provided by the inversion with constraints from MOPITTv8-NIR-TIR from the 1$^{st}$ to the 7$^{th}$, March 2015, in %.*

| prior | | | | posterior | | | |
|---|---|---|---|---|---|---|---|
| MB | RMSE | STD | r | MB | RMSE | STD | r |
| 15.88 | 41.95 | 38.82 | 0.02 (p value = 0.99) | 11.58 | 21.14 | 17.69 | 0.74 (p value = 2.08x10$^{-11}$) |

**Table 4.** *Statistics for the comparison between simulated and observed CO surface concentrations over Central and Eastern Europe (see the area in purple in Figure 5). MB= Mean Bias, RMSE= Root Mean Square Error, STD= Standard Deviation are in ppbv. The spatial correlations r are presented with their p value.*

### 4.2.4. Inversion of NO$_x$ emissions

The prior NO$_x$ emissions and the corrections provided by the different sensitivity tests of Table 3 are shown in Figure 8. Here, we analyzed the results from inversion E. As expected from the underestimation of the prior tropospheric columns in Figure 6a, local increments may be large, for example over industrial areas (e.g., over the Po Valley) and over the Netherlands, with increments of more than +50% (Figure 8b). The analyzed NO$_2$ tropospheric columns in Figure 6c are the columns simulated by CHIMERE with the NO$_2$ posterior fluxes: as expected, the optimization of the fluxes improves the fit of the simulated concentrations to the observations over the Netherlands (Figure 6e). Over this area (see the purple box in Figure 6), where the OMI uncertainties are lower

than 50% (Figure 9b), the mean bias between the simulation and the observations has been reduced by about 24% when using the posterior emissions (mean bias of $1.9 \times 10^{15}$ molec.cm$^{-2}$ against $2.6 \times 10^{15}$ molec.cm$^{-2}$ with the prior emissions, Table 5, Figure 9a). The RMSE and the standard deviation have been reduced by about 7%. The correlation has not been improved.

Even with high emission increments, the impact on the tropospheric columns is rather small. We have performed a test to explain this lack of sensitivity. We have simulated $NO_2$ columns with anthropogenic emissions increased by a factor 3 compared to the simulation in Figure 6a. The ratio between these two simulations shows strong non-linearities, blurring the multiplicative effect of our increments and explaining the lack of sensitivity (not shown). By increasing $NO_x$ anthropogenic emissions, $NO_2$ tropospheric columns can be strongly increased and can even exceed the observations values for particular pixels. $NO_2$ tropospheric columns can also be decreased or only slightly increased. On average, it tends to increase the concentrations by a factor that is much smaller than the factor of increase in the anthropogenic emissions. However, the patterns where the posterior tropospheric columns exceed the observations or, on the opposite are decreased or only slightly increased, explain why the inversion system does not attempt at increasing further the average level of the concentration (to decrease further the general bias to the observations), even though it accounts for the impact of non-linearities in the chemistry through the use of the M1QN3 minimization algorithm. We can conclude that the strong non-linearities of the $NO_x$ chemistry mainly explain the lack of sensitivity between $NO_x$ emissions and satellite $NO_2$ columns. Besides chemical effects, the lack of sensitivity could be also partly due to the contribution of emissions during the preceding days and the assimilation window will be widened in the near future.

The posterior emissions and their uncertainties will have to be evaluated and may bring hints to the cause of the discrepancies between simulated and observed $NO_2$ tropospheric columns. The biases between OMI and simulated $NO_2$ tropospheric columns are a complex topic that is not related to our CHIMERE simulations only [Huijnen et al., 2010; Souri et al., 2020; Elguindi et al., 2020]. Several studies have indeed already reported that strong non-linear relationships exist between $NO_x$ emissions and satellite $NO_2$ columns [Lamsal et al., 2011; Vinken et al., 2014; Miyazaki et al., 2017; Li and Wang, 2019]. This reveals that a fully comprehensive scientific study is required, by analyzing the $NO_x$ lifetime through processes such as the $NO_2+OH$ reactions and/or the reactive uptake of $NO_2$ and $N_2O_5$ by aerosols [e.g. Lin et al., 2012; Stavrakou et al., 2013].

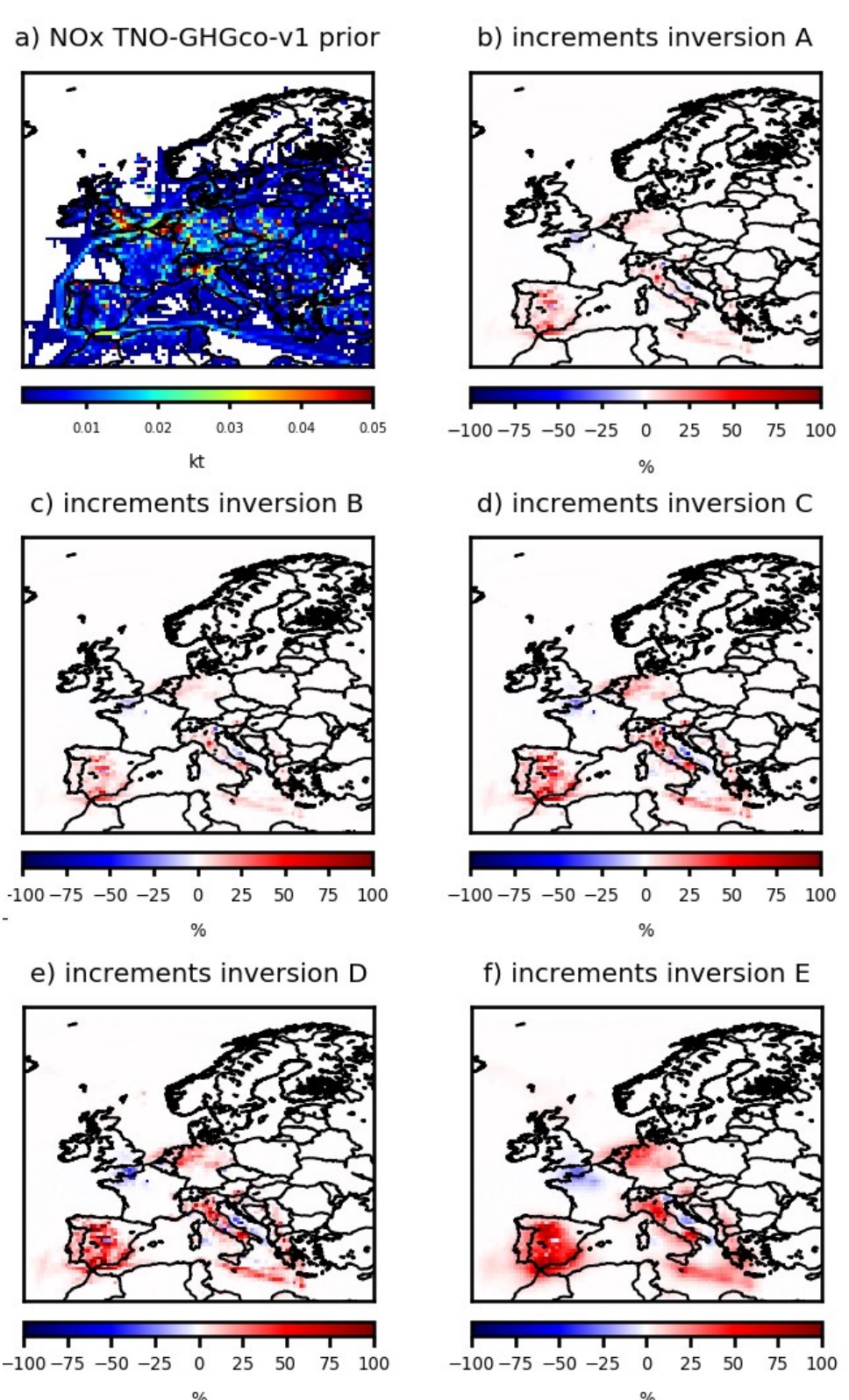

632

**Figure 8.** *a) TNO-GHGco-v1 $NO_x$ anthropogenic prior emissions, in $ktNO_2$/grid-cell and*
633
*increments provided by the inversion b) A, c) B, d) C, e) D and f) E with constraints from OMI the*
634
*19[th], February 2015, in %. The description of the different inversions is given in Table 3.*
635

636

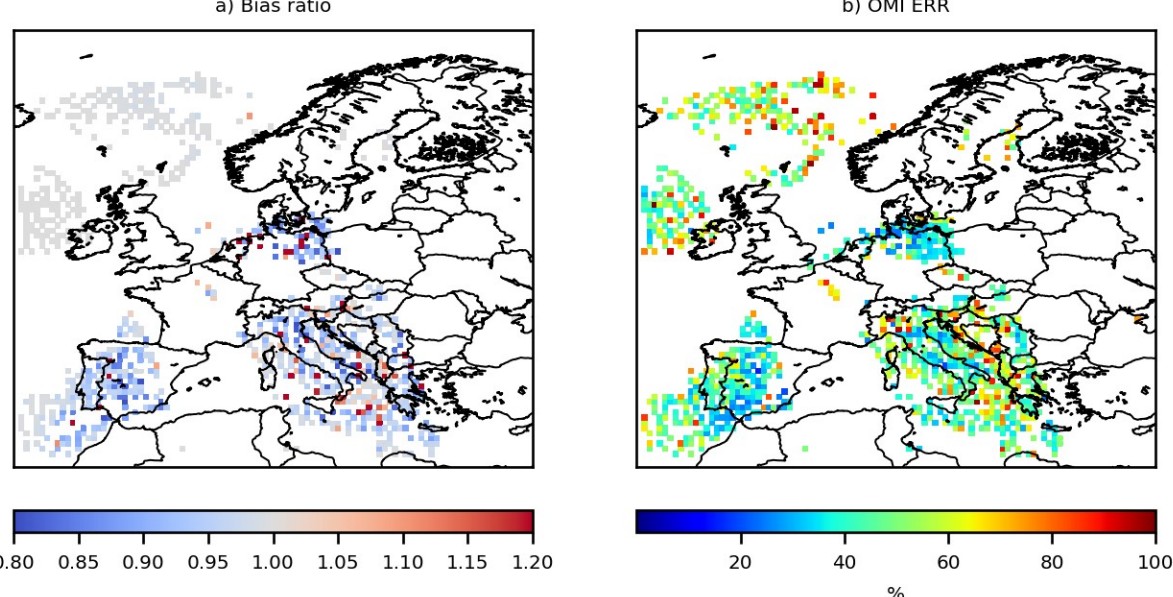

637

**Figure 9**. *a) Bias ratio between CHIMERE simulations using the posterior emissions against prior TNO-GHGco-v1 emissions compared to the OMI-QA4ECV-v1.1 observations. All ratios lower than 1, in blue, demonstrate that posterior emissions improve the simulation compared to the prior ones. b) OMI uncertainties, in %, the 19[th], February 2015.*

642

| | prior | | | | posterior | | | |
|---|---|---|---|---|---|---|---|---|
| | MB | RMSE | STD | r | MB | RMSE | STD | r |
| NO$_2$ | $2.6 \times 10^{15}$ | $4.0 \times 10^{15}$ | $3.0 \times 10^{15}$ | 0.008 (p=0.96) | $1.9 \times 10^{15}$ | $3.74 \times 10^{15}$ | $2.9 \times 10^{15}$ | 0.01 (p=0.91) |

**Table 5.** *Statistics for the comparison between simulated and observed NO$_2$ tropospheric columns for the inversion E, mainly over the Netherlands (see the area in purple in Figure 6). MB= Mean Bias, RMSE= Root Mean Square Error, STD= Standard Deviation are in molecm.cm$^{-2}$. The spatial correlations r are presented with their p value.*

**5. Conclusion/Discussion**

This paper presents the Bayesian variational inverse system PYVAR-CHIMERE, which has been adapted to the inversion of reactive species such as CO and NO$_x$, taking advantage of the previous developments for long-lived species such as CO$_2$ [Broquet et al., 2011] and CH$_4$ [Pison et al., 2018]. We show the potential of PYVAR-CHIMERE, with inversions for CO and NO$_x$ illustrated over Europe. PYVAR-CHIMERE will now be used to infer CO and NO$_x$ emissions over long periods, e.g. first for a whole season or year and then for the recent decade 2005-2015 in the framework of the H2020 VERIFY project over Europe, and in the framework of the ANR PolEASIA project over China, to quantify their trend and their spatio-temporal variability. Nevertheless, as we have reported strong non-linear relationships between NO$_x$ emissions and

satellite $NO_2$ columns, a fully comprehensive scientific study is required, by analyzing the $NO_x$ lifetime through processes such as the $NO_2$+OH reactions and/or the reactive uptake of $NO_2$ and $N_2O_5$ by aerosols [e.g. Lin et al., 2012; Stavrakou et al., 2013]. Biogenic emissions will be also further studied to better understand the relationship between $NO_x$ emissions and $NO_2$ spaceborne columns.

The PYVAR-CHIMERE system can handle any large number of both control parameters and observations. It will be able to cope with the dramatic increase in the number of data in the near future with, for example, the high-resolution imaging (pixel of 7x3.5 $km^2$) of the new Sentinel-5P/TROPOMI program, launched in October 2017. These new space missions with high-resolution imaging have the ambition to monitor atmospheric chemical composition for the quantification of anthropogenic emissions. It will indeed entail using PYVAR-CHIMERE at higher spatio-temporal resolutions, but probably for smaller domains (i.e., over countries rather than over Europe) as a compromise between resolution and the computational cost. Moreover, a step forward in the joint assimilation of co-emitted pollutants will be possible with the PYVAR-CHIMERE system and the availability of TROPOMI co-localized images of CO and $NO_2$. This should improve the consistency of the inversion results and can be used to inform inventory compilers, and subsequently improve emission inventories. Moreover, this development will help in further understanding air quality problems and addressing air quality related emissions at the national to subnational scales.

**Author Contribution**
All authors have contributed to the manuscript writing (main authors: AFC, GB, IP and GD) and to the development of the present version of the PYVAR-CHIMERE system (main developer: IP). IP and GD have parallelized the adjoint version from Menut et al., [2000], Menut et al., [2003] and Pison et al., [2007]. IP has complemented the adjoint of new parameterizations since the CHIMERE release in 2011 and the tangent-linear model.

**Code and Data Availability**
OMI QA4ECV $NO_2$ product can be found here: http://temis.nl/qa4ecv/no2.html.
MOPITTv8-NIR-TIR CO product can be found here: ftp://l5ftl01.larc.nasa.gov/MOPITT/
The CHIMERE code is available here: www.lmd.polytechnique.fr/chimere/.

The associated documentation of PYVAR-CHIMERE is available on the website https://pyvar.lsce.ipsl.fr/doku.php/3chimere:headpage. The documentation includes a whole description of PYVAR-CHIMERE and several tutorials on how to run a first PYVAR-CHIMERE simulation or how to run an inversion.

**Competing interests**
The authors declare that they have no conflict of interest.

**Acknowledgements**

We acknowledge L. Menut and C. Schmechtig for their contributions to the development work on the adjoint code of CHIMERE and its parallelization. We acknowledge the TNO team (H.A. Denier van der Gon, J. Kuenen, S. Dellaert, S.Jonkers, A. Visschedijk, et al.) for providing $NO_x$ and CO emissions over Europe.We also acknowledge the free use of tropospheric $NO_2$ column data from the OMI sensor from http://temis.nl/qa4ecv/no2.htmland the free use of CO surface concentrations from the MOPITT sensor from ftp://l5ftl01.larc.nasa.gov/MOPITT/. For this study, A. Fortems-Cheiney was funded by the French Space Agency-Centre National d'Etudes Spatiales CNES and by the H2020 VERIFY project, funded by the European Commission Horizon 2020 research and innovation programme, under agreement number 776810. L. Costantino was funded by the PolEASIA ANR project under the allocation ANR-15-CE04-0005. This work was granted access to the HPC resources of TGCC under the allocations A0050107232 and A0070102201 made by GENCI. Finally, we wish to thank F. Marabelle (LSCE) and his team for computer support.

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
