# Peer review of "Variational regional inverse modeling of reactive species emissions"

_Geoscientific Model Development, 2019_

## Referee Comment (RC1) · Anonymous Referee #1 · 9 Sep 2019

This paper describes a Bayesian inverse modeling system for reactive compounds, PYVAR-CHIMERE. It also provides an illustration of what this system can do with two one-day inversions of emissions over Europe. The paper is generally well written, and the topic of the paper is relevant for this journal. Although the results indicate that the system has potential, there are several major issues which should be dealt with before this work can be published in GMD.

1) poor quality of the figures and equations. On the screen, it is more or less acceptable (with the zoom in feature of my pdf viewer), but upon printing, many figures (especially Figs. 1-4 and Figs. 6a and 8a) and equations are impossible to read. Figure 3 is really

a Table and should be inserted as such. Some mathematical symbols (e.g. gradient on line 158, multiplication on line 167) are inappropriate.

2) Section 3.2 is 'Development of the adjoint of CHIMERE'. But the adjoint of CHIMERE was developed a long time ago (publications by L. Menut, I. Pison). What are the specific developments realized for this study, besides the minor changes to CHIMERE mentioned in the text ?

3) the results of inverse modeling studies are very dependent on the inversion setup, in particular the definition of the control vector and the construction of the covariance matrices **R** and especially **B**. What is the strategy in this regard? I understand that the main purpose of the paper is less to present specific inversion studies than to describe the general modeling framework. But the very simplistic choices made for the two one-day inversions suggest an absence of any strategy. The chemical lifetimes of the target species, the duration of the experiments, the initial and boundary conditions, and the assumed a priori uncertainties should all play a role in the inverse setup definition.

4) the two illustrations of PYVAR-CHIMERE capabilities are unconvincing. Yes, the system finds a minimum to the cost function, and the a posteriori simulation matches the observations quite well; but no, the a posteriori emissions are not shown to be closer to reality. With its long lifetime, CO is largely determined by the initial and lateral boundary conditions, which are part of the control state vector being optimized. The paper does not provide information on the a priori uncertainties for these parameters. A discussion is needed, and possibly sensitivity simulations. Note that, although the a priori simulation overestimates CO over Central Europe (south of Poland), the inversion increases the emissions there by about a factor of 2! Over Germany, the emissions are almost doubled in the Southern part, but are unchanged elsewhere. How can this be justified? Even for NOx, in spite of their shorter lifetime, the initial and boundary conditions play probably a very important role, as indicated by the strong reduction of the model biases over ocean (compare Fig. 7c and 7d). The discussion of the results for NOx (lines 373-376) is impossible to understand. It says that the optimization of

[Figure]

NOx fluxes has only a small impact on the model biases. This is not true. Comparison of Fig. 7c and 7d show that the optimization works very well ! The authors claim that PYVAR optimizes only the NO2 fluxes, not those of NO. I don't believe this, it doesn't make sense. Please check this. In any case, clarifications and possibly a sensitivity analysis are in order.

Other comments:

- throughout the text, replace "NO2 emissions" by "NOx emissions" (if indeed, as should be the case, NOx emissions are optimized, not just NO2)

- lines 107-110: please refer also to GEOS-Chem adjoint papers (Henze, Kopacz, Cao etc.)

- l. 149 what is meant by "the control of emissions"? Please rephrase

- l. 168-169 What are the thresholds for the ratio between final and initial gradient norm, and for the number of iterations?

- Figure 1 In the orange box on the right, the order of operators should go backwards, shouldn't they?

- l. 195 leads (instead of lead)

- Figure 3. Explain the meaning of the "correction type" and of the three numbers in column "B variance coefficients".

- Figure 4. The legends mentions text in blue and in grey. I don't see that on the figure. Is this figure useful?

- l. 298 What is the resolution of ECMWF data? Are those data interpolated to the model grid?

- l. 300 Derognat et al. 2003 does not present a chemical mechanism, but refers to earlier papers.

- What are the a priori lateral boundary conditions?

- on l. 303 and legend of Fig. 8, the information required to run the inversion are said to be listed in Table 1. This is not correct.

- l. 304 Are only anthropogenic emissions optimized? Or the total of all emissions?

- l. 306-307 The 3D initial conditions at the model resolution are said to represent 8585 components of the control vector. But what about the vertical dependence?

- l. 311-314 What are the non-anthropogenic emissions used in the model? Please provide a webpage and reference for EMEP emissions. Aren't there any publication or webpage for the TNO emissions?

- l. 328 Why the median?

- l. 341 With errors of 100% on the emissions, how can negative a posteriori emissions be avoided? How is this dealt with?

- Section 4.2 The Figure 5 shows both underestimations and overestimations by the a priori simulation. This is not well reflected in the discussion.

- Figure 7c and 7d should show absolute differences. A better color scale should be possible for Fig. 7a and 7b.

―――――――――――――――

---

## Short Comment (SC1) · 5 Nov 2019

Dear authors,

in my role as Executive editor of GMD, I would like to bring to your attention our Editorial version 1.2:

https://www.geosci-model-dev.net/12/2215/2019/

This highlights some requirements of papers published in GMD, which is also available on the GMD website in the 'Manuscript Types' section:

http://www.geoscientific-model-development.net/submission/manuscript_types.html

In particular, please note that for your paper, the following requirements have not been met in the Discussions paper:

- The main paper must give the model name and version number (or other unique identifier) in the title.

Therefore please provide the version number of the PYVAR-CHIMERE which is actually used for the current publication in the title in your revised submission to GMD.

Yours,

Astrid Kerkweg

---

## Referee Comment (RC2) · Anonymous Referee #2 · 5 Dec 2019

The paper describes the variational data assimilation version of the CHIMERE, PYVAR-CHIMERE, which is capable of inversions of reactive gases. As a demonstration to the newly developed code, the inversion of CO and NO2 is shown for two different days in late winter/early spring 2015. The papers topic is of good relevance for GMD and contribute to a documented open source regional data assimilation system for reactive chemistry. Although the paper is generally well written, major changes are requested before publishing the manuscript in GMD. These major changes are:

- The quality of figures and formulas is unacceptable. Arrows should be larger/thicker (Fig. 1 and 2), annotations in Figs. 5, 6, 7, and 8 are too small, separation of subplots

in Figs. 5 and 7 should be clearer. - The description of the inversion is unsatisfactory. The cost function and its gradient should explicitly show the model operator M, which is currently included in the state vector x. Further, it is unclear how the emissions are corrected. How can negative emissions be avoided? Are the emissions optimized for each time step or for the whole assimilation window? Are the emissions constant for the simulation time or does the inversion result in correction factors for the emissions. Then, the special treatment of 4D-var for emission factor optimization should be shown, e. g. how the positive definiteness of the correction factors is ensured. The manuscript must be more precise in this context.

- The calculation of the size of the control vector is erroneous. The vertical dependence of the initial conditions is missing in the calculation.

- In the experiment section (section 4) no information on the initial and boundary conditions is given. It should be illustrated to what degree both are changed during the inversion. Further, a comparison or sensitivity test should be shown on what the impact of emission optimization is compared to a joint optimization with initial and boundary conditions.

- Although the two test cases show a reduction of the difference between the assimilated observation and the analysis, this is not a proof of the successful operation of the data assimilation algorithm. A comparison with independent observations and a table with quality measures (e. g. bias, root mean square error, cost reduction) is necessary. It is advised to perform the analysis on a few consecutive days to assess the stability and quality of the inversion on different days.

- in the description of the test cases both, initial values and boundary conditions are included in the control vector, thus, the analysis is not complete without showing these two variables. A discussion is needed about the correction for all three variables, i. e. emissions, initial values, and boundary conditions, their relative influence on the analysis and about potential limitations of the inversion.

- If the full adjoint of the chemical processes is used there should be an adjoint signal for other species than CO and NO2 as well. This must be clarified. Are these signals simply not considered or not discussed? What is the reason for not optimizing NO emissions then?

- the model resolution of 0.5 x 0.5 square degrees seems to be a bit coarse for anthropogenic emission assessments. Is nesting available? A discussion on this point is needed.

- a better description of the B-matrix is needed in section 4.1.3. What about correlations for initial and boundary conditions?

Further minor coments:

- line 30: (VOCs) instead of "(VOCs))"

- line 39: reference for (LRTAP) would be appreciated

- line 43: no commas

- line 85: CO and NOx (instead of "CO, NOx")

- line 88: citation van der A. [2008] is not appropriate (van der A et al. [2008]), also in the reminder of the manuscript

- line 93/94: ".. for which variational methods are more suitable than KFs by design": a reference would be appreciated for this statement.

- line 122/123: of the current inversion (instead "of the inversion")

- line 163: quasi-Newton (instead "quasi-Newtonian")

- line 165: Reference for incremental 4D-var approach is appreciated

- line 203: It would be appreciated if the manuscript contains a table with the available (and adjoint) processes of CHIMERE

- line 227/228: better: "PVAR, CHIMERE, and text sources are displayed in blue, orange, and grey boxes, respectively."

- caption of figure 4: better: "Simplified scheme of how PYVAR scripts prepare the observations y using satellite data. PYVAR and text sources are displayed in blue and grey boxes, respectively."

- line 264: Equation "Cm =" is not a correct mathematical formulation, Cm(o) is a column, xa is the state vector (a profile in this context).

- line 284/285 (general remark): The models speedup increases enormously if the master process is additionally used for calculation and parallel IO is included for suitable IO operations.

- line 290: ... days for CO and NO2, respectively (instead of "... days, respectively for CO and NO2")

- line 302: Table 1 is not control vector specific. This sentence can be removed

- line 304: for one day (instead "at a 1-day"); resolution (instead "resolutions")

- line 313/314: a spinup for the initial values is needed for an appropriate analysis, otherwise the model may be to far off the observations for a suitable correction

- line 317: Reference for MOPITT is missing

- line 328: MOPITT instead of "OMI"

- page 12, line 4: flown instead of "flying"

- line 368: parts (instead of "part"); present (instead of "presents")

- page 14, last line: particularly over the Po Valley (instead ", and particularly over Po Valley")

- caption Fig. 7 d: is it really the difference between prior and posterior? Inconsistency with text (see next point)

**GMDD**

- line 374: Fig. 5c seems to be wrong here. Is it Fig. 7d?

- line 380/381: Using the full adjoint of CHIMERE, this must already be available. Please check for adjoint NO signals

- line 399: remove "for example"

---

## Author Comment (AC1) · 29 Apr 2020

**We wish to thank the referee for his/her helpful comments. The full reviews are copied hereafter and our responses are inserted. The comments of the reviewer are in normal black and our answers in bold.**

This paper describes a Bayesian inverse modeling system for reactive compounds, PYVAR-CHIMERE. It also provides an illustration of what this system can do with two one-day inversions of emissions over Europe. The paper is generally well written, and the topic of the paper is relevant for this journal. Although the results indicate that the system has potential, there are several major issues which should be dealt with before this work can be published in GMD.

1) poor quality of the figures and equations. On the screen, it is more or less acceptable (with the zoom in feature of my pdf viewer), but upon printing, many figures (especially Figs. 1-4 and Figs. 6a and 8a) and equations are impossible to read.
**We apologize, the poor quality in particular of the equations was due to conversion from OpenOffice to pdf. The resolution of the equations and of the figures has been improved.**

Figure 3 is really a Table and should be inserted as such.
**We agree, Figure 3 is now the Table 1.**

Some mathematical symbols (e.g. gradient on line 158, multiplication on line 167) are inappropriate.
**These symbols have been corrected.**

2) Section 3.2 is 'Development of the adjoint of CHIMERE'. But the adjoint of CHIMERE was developed a long time ago (publications by L. Menut, I. Pison). What are the specific developments realized for this study, besides the minor changes to CHIMERE mentioned in the text?
**Indeed, the adjoint of CHIMERE was developed a long time ago. We first have changed the title section into "Development and parallelization of the adjoint and tangent-linear codes of CHIMERE".**

**We added sentences in Section 3.1: "PYVAR has been adapted to CHIMERE with an adjoint code without chemistry a first time by Broquet et al. [2011]. In order to couple PYVAR to the new state-of-the-art version of CHIMERE (see Section 3.2), to include chemistry, and to increase its modularity, flexibility and clarity, the new system described here has been developed. It includes elements of the inversion system (coded in Fortran90) of [Pison et al., 2007]."**

**Efforts have indeed been made for the parallelization of the code. This is now explained in the text in Section 3.2: "Then, it has been parallelized at LSCE and LISA. This work required a redesigning of the whole code, associated with a full testing scheme. Furthermore, the tangent-linear (TL) code has been developed and validated at LSCE. Changes have been implemented in the forward CHIMERE code embedded in PYVAR-CHIMERE to match requirements of the studies lead with PYVAR-CHIMERE. These changes have been implemented in both the adjoint and the TL codes. Compared to the CHIMERE 2013 version [Menut et al., 2013], the most important of these changes are:**
**•For the geometry, the possibility of polar domains and the use of the coordinates of the corners of the cells instead of only the centers**
**•For the transport, the non-uniform Van Leer transport scheme on the horizontal,**

**•For chemistry, various switches have been added to avoid going into the chemistry, deposition and wet deposition routines when no species requires them (e.g. no chemistry for methane at a regional scale)."**

3) the results of inverse modeling studies are very dependent on the inversion setup, in particular the definition of the control vector and the construction of the covariance matrices R and especially B. What is the strategy in this regard? I understand that the main purpose of the paper is less to present specific inversion studies than to describe the general modeling framework. But the very simplistic choices made for the two one-day inversions suggest an absence of any strategy. The chemical lifetimes of the target species, the duration of the experiments, the initial and boundary conditions, and the assumed a priori uncertainties should all play a role in the inverse setup definition.

**Indeed, we agree, the main purpose of a GMD paper is less to present specific inversion studies than to describe the general modeling framework. Nevertheless, we added information to explain our choices of illustrations in the introduction of Section 4: "We have chosen to present illustration of CO inversion over a 7-day window, the first week of March 2015. Considering the short lifetime of $NO_x$ of a few hours [Valin et al., 2013; Liu et al., 2016], we have chosen to present illustration of $NO_x$ inversion over a 1-day window, the 19th February 2015. These particular periods have been chosen as they present a representative number of super-observations during winter, and as the emissions are high during that period."**

**We also have added a new "Section 4.2.2. Covariance matrices B and R" to better describe the covariance matrices B and R.**

4) the two illustrations of PYVAR-CHIMERE capabilities are unconvincing. Yes, the system finds a minimum to the cost function, and the a posteriori simulation matches the observations quite well; but no, the a posteriori emissions are not shown to be closer to reality. With its long lifetime, CO is largely determined by the initial and lateral boundary conditions, which are part of the control state vector being optimized. The paper does not provide information on the a priori uncertainties for these parameters. A discussion is needed, and possibly sensitivity simulations.

**We now provide more information about the initial, lateral and top boundary conditions in the new "Section 4.1.2. CHIMERE set-up": "Different climatological values from the LMDZ-INCA global model [Szopa et al., 2008] or from a MACC reanalysis are used to prescribe concentrations at the lateral and top boundaries and the initial atmospheric composition in the domain."**

**Information on the prior uncertainties was given in the former Figure 3, now replaced by Table 1. We also added sensitivity tests in Section 4.1.3: "With its lifetime of about two months, CO could be strongly driven by the initial and lateral boundary conditions prescribed in the CTM. In fact, as seen in Figure 4b, initial and boundary conditions provide a relatively flat background and the patterns which appear clearly over the background are linked to surface emissions (Figure 4a). To characterize the uncertainties in the concentration fields due to the initial and lateral boundary conditions, we performed a sensitivity test by using either climatological values from LMDZ-INCA or a MACC reanalysis: the results were not significantly different, with relative differences in concentrations of less than 15% over continental land (Figure 5c)."**

**We also added text in the new "Section 4.2.2. Covariance matrices B and R": "Based on the sensitivity test in Figure 5, the errors assigned to the CO lateral and top boundary conditions and to their initial conditions are set at 15%. As these relative errors are significantly lower than those for the emissions and as variations in the CO surface concentrations are mainly**

**driven by emissions (Figure 4), we assume a small relative influence of the correction of initial and boundary conditions on our results. "**

Note that, although the a priori simulation overestimates CO over Central Europe (south of Poland), the inversion increases the emissions there by about a factor of 2! Over Germany, the emissions are almost doubled in the Southern part, but are unchanged elsewhere. How can this be justified?
**Figure 6 has been updated. The emissions are increased over Central and Eastern Europe, except in the south of Poland.**

Even for NOx, in spite of their shorter lifetime, the initial and boundary conditions play probably a very important role.
**We also checked the impact of initial and boundary conditions on NO$_2$ tropospheric columns. Due to its short lifetime, the impact is even smaller than for CO. We chose not to show this sensitivity test in the paper.**

The discussion of the results for NOx (lines 373-376) is impossible to understand. It says that the optimization of NOx fluxes has only a small impact on the model biases. This is not true. Comparison of Fig. 7c and 7d show that the optimization works very well!
**We do not agree, as Fig 7c and Fig 7d were not comparable (they did not have the same legend). Nevertheless, we agree, it could have been confusing. We now present the impact of the optimization of the NO$_x$ fluxes differently, in Figure 7.**

The authors claim that PYVAR optimizes only the NO2 fluxes, not those of NO. I don't believe this, it doesn't make sense. Please check this. In any case, clarifications and possibly a sensitivity analysis are in order.
**As both the reviewers have been disturbed by the illustration with only NO$_2$ fluxes, we now present inversion for NO$_x$ emissions.**

Other comments:
- throughout the text, replace "NO2 emissions" by "NOx emissions" (if indeed, as should be the case, NOx emissions are optimized, not just NO2)
**See comments above. "NO$_2$ emissions" have indeed been replaced by "NO$_x$" emissions throughout the text.**

- lines 107-110: please refer also to GEOS-Chem adjoint papers (Henze,Kopacz, Caoetc.)
**A reference to Henze et al. 2007 has been added. A reference to the adjoint-based four-dimensional variational (4D-Var) assimilation system, WRF-CO2 4D-Var, has also been added.**

- l. 149 what is meant by "the control of emissions"?
**We have rephrased: "By definition, the observation errors combine errors in both the data and the observation operator, in particular measurement errors and errors in the conversion of satellite measurement into concentration data, errors from the CTM, representativity errors due to the comparison between point measurements and gridded models or due to the representation of the fluxes as gridded maps at a given spatial resolution, and aggregation errors associated with the optimization of emissions at a given spatial and/or temporal resolution (as specified in the control vector) that is different from (usually coarser than) that of the CTM [Wang et al., 2017]."**

-l. 168-169 What are the thresholds for the ratio between final and initial gradient norm, and for the number of iterations?

This information was already given in Section 4. It is now in the introduction of Section 4: "For practical purposes, we recommend to reduce the norm of the gradient of $J$ by 90%. We no longer give information about the number of iterations as it depends on each configuration system. These information are now given in Section 4.2.3 for CO: " Ten iterations are needed to reduce the norm of the gradient of $J$ by 90% with the minimization algorithm M1QN3" and in Section 4.2.4 for $NO_x$: "Six iterations are needed to reduce the norm of the gradient of $J$ by 85% with the minimization algorithm M1QN3".

-Figure 1. In the orange box on the right, the order of operators should go backwards, shouldn't they?
Indeed, the adjoint go backwards but these operations are made simultaneously in CHIMERE.

- l. 195 leads (instead of lead)
It has been changed.

-Figure 3. Explain the meaning of the "correction type" and of the three numbers in column "B variance coefficients".
The correction type describes the way the emissions are corrected by the inversion. We have added the following descriptions in "Section 3.3. Definition of the control vector": "Several types of corrections can be applied, they are defined in the code as "add", "mult" or "scale". Both the corrections "add" and "mult" are applied to gridded control variables. For correction type "add" the control variables are increments added to the corresponding components of the model inputs. For correction type "mult", the control variables are scaling factors multiplying the corresponding components of the model inputs. The difference between the two options "add" and "mult" plays a role when inverting fluxes which can switch from positive to negative values (like $CO_2$ natural fluxes). For type "scale", the corrections consist in applying scaling factors to activity maps and/or masks for regions (which is similar to the control of budgets for different regions, types of activities, and/or processes in inversions where the control vector is not gridded [Wang et al., 2018]) and adding the obtained values to the corresponding components of the model inputs."

The three numbers in column "B variance coefficients" are standard deviation coefficient. We have added information: "The variances are specified by the user through standard deviation coefficient (Table 1), which can be a fixed value ("fx") or a percentage ("pc") to define the diagonal standard deviation matrix $\sum$."

- Figure 4. The legends mentions text in blue and in grey. I don't see that on the figure. Is this figure useful?
The sentence mentioning text in blue and in grey has been removed.

- l. 298 What is the resolution of ECMWF data? Are those data interpolated to the model grid?
The spatial resolution of the ECMWF data is 0.25°x0.25°. They have been interpolated to the model grid.

-l. 300 Derognat et al. 2003 does not present a chemical mechanism, but refers to earlier papers.
The reference to Derognat has been removed. We now refer to Lattuati, 1997 and to the lastest CHIMERE documentation. The text is now:"The chemical scheme used in PYVAR-CHIMERE is MELCHIOR-2, with more than 100 reactions [Lattuati, 1997; CHIMERE 2017], including 24 for inorganic chemistry".

-What are the a priori lateral boundary conditions?

**The prior lateral and top boundary conditions are climatological values from the LMDZ-INCA global model [Szopa et al., 2008]. It was indicated in Section 4.1.1. We now describe them in "Section 4.1.2 CHIMERE set-up" and we also made a sensitivity test with a MACC reanalysis in "Section 4.1.3. Sensitivity to emissions and to initial and boundary conditions".**

- on l. 303 and legend of Fig. 8, the information required to run the inversion are said to be listed in Table 1. This is not correct.
**We agree, this is now true (Figure 3 is now called Table 1).**

- l. 304 Are only anthropogenic emissions optimized? Or the total of all emissions?
**Only the anthropogenic emissions are optimized here. This is now written in the introduction of Section 4: "The potential of the PYVAR-CHIMERE system to invert emissions of reactive species is illustrated with the inversion of CO and NO$_x$ anthropogenic emissions in Europe respectively based on MOPITT CO data and OMI NO$_2$ data".**

**This is also now written in the description of the control vectors in Section 4.2.1.**

-l. 306-307 The 3D initial conditions at the model resolution are said to represent 8585components of the control vector. But what about the vertical dependence?
**The vertical dependence is indeed taken into account. The number of components in the control vector has been corrected.**

-l. 311-314 What are the non-anthropogenic emissions used in the model?
**The biogenic emissions, that we assume negligible in winter, are not used in our illustration. In addition, we should have described that PYVAR-CHIMERE only infers anthropogenic emissions at this stage. This is now added in Section3.2 : "It should also be noted that PYVAR-CHIMERE only infer anthropogenic emissions at this stage. The optimization of biogenic emissions, which are linearly interpolated at the sub-hourly scale in CHIMERE, is currently under development."**

-Please provide a webpage and reference for EMEP emissions. Aren't there any publication or webpage for the TNO emissions?
**We now provide two references for the EMEP emissions (a publication and a webpage). We also provide a reference for the TNO-GHGco used in this paper (submitted in December 2019 and published in February 2020). The text is now: "The prior anthropogenic emissions for CO and NO$_x$ emissions come from the TNO-GCHco-v1inventory [Super et al., 2020], the last update of the TNO-MACCII inventory [Kuenen et al., 2014]. The prior anthropogenic emissions for VOCs come from the EMEP inventory [Vestreng et al., 2005; EMEP/CEIP website]."**

-l. 328 Why the median?
**The median is chosen here to take proper account of the AKs (we can not take the mean of the AKs).**

-l. 341 With errors of 100% on the emissions, how can negative a posteriori emissions be avoided? How is this dealt with?
**We now answered to this remark in "Section 4.2.2. Covariance matrices B and R": "With such a set-up, in theory, we could obtained negative posterior emissions since the inversion system does not impose a constraint of positivity in the results. Nevertheless, even 100% of uncertainty lead to a prior distribution mostly (>80%) on the positive side. The assimilation of data showing an increase above the background (at the edges of the domain; not shown)**

**further drive the inversion towards positive emissions for both CO and NO$_x$ inversions. In practice, our inversion does not lead to negative posterior emissions (Figure 7b). Spatial and temporal correlations in B would further limit the probability to get negative emissions locally by smoothing the posterior emissions at a spatial scale at which the "aggregated" prior uncertainty is smaller than 100%. However, a positivity constraint should be implemented in future versions of the system."**

-Section 4.2 The Figure 5 shows both underestimations and overestimations by the apriori simulation. This is not well reflected in the discussion.
**Indeed, this is now reflected in the discussion in Section 4.2.3.**

-Figure 7c and 7d should show absolute differences.
**We no longer present these figures.**

-A better color scale should be possible for Fig. 7a and 7c
**Indeed, it has been done in Figure 8 and in Figure 10.**

---

## Author Comment (AC2) · 29 Apr 2020

**We wish to thank the referee for his/her helpful comments. The full reviews are copied hereafter and our responses are inserted. The comments of the reviewer are in normal black and our answers in bold.**

The paper describes the variational data assimilation version of the CHIMERE,PYVAR-CHIMERE, which is capable of inversions of reactive gases. As a demonstration to the newly developed code, the inversion of CO and NO2 is shown for two different days in late winter/early spring 2015. The papers topic is of good relevance for GMD and contribute to a documented open source regional data assimilation system for reactive chemistry. Although the paper is generally well written, major changes are requested before publishing the manuscript in GMD.

These major changes are:
- The quality of figures and formulas is unacceptable. Arrows should be larger/thicker
(Fig. 1 and 2), annotations in Figs. 5, 6, 7, and 8 are too small, separation of subplots and 7 should be clearer.
**The quality of the all the figures have been improved.**

- The description of the inversion is unsatisfactory. The cost function and its gradient should explicitly show the model operator M, which is currently included in the state vector x.
**We do not agree with this statement, the model operator is not included in the state vector x.**

- Further, it is unclear how the emissions are corrected. How can negative emissions be avoided?
**We now answered to this remark in "Section 4.2.2. Covariance matrices B and R": "With such a set-up, in theory, we could obtained negative posterior emissions since the inversion system does not impose a constraint of positivity in the results. Nevertheless, even 100% of uncertainty lead to a prior distribution mostly (>80%) on the positive side. The assimilation of data showing an increase above the background (at the edges of the domain; not shown) further drive the inversion towards positive emissions for both CO and NO$_x$ inversions. In practice, our inversion does not lead to negative posterior emissions (Figure 7b). Spatial and temporal correlations in B would further limit the probability to get negative emissions locally by smoothing the posterior emissions at a spatial scale at which the "aggregated" prior uncertainty is smaller than 100%. However, a positivity constraint should be implemented in future versions of the system."**

- Are the emissions optimized for each time step or for the whole assimilation window?
**The user can chose the time resolution at which the emissions are optimized. In our illustrations, we now present inversions at 7-day and at 1-day resolutions.**

-Are the emissions constant for the simulation time or does the inversion result in correction factors for the emissions?
**Indeed, the emissions are inverted, i.e, the inversion results in correction factors for the emissions at the specified time and spatial scales.**

-Then, the special treatment of 4D-var for emission factor optimization should be shown, e. g. how the positive definiteness of the correction factors is ensured. The manuscript must be more precise in this context.
**We do not agree. We would like to emphasize that the PYVAR-CHIMERE system for inversion is not a 4D-VAR one.**

-The calculation of the size of the control vector is erroneous. The vertical dependence of the initial conditions is missing in the calculation.
**The vertical dependence is indeed taken into account. The number of components in the control vector has been corrected.**

-In the experiment section (section 4) no information on the initial and boundary conditions is given.
**We now provide more information about the initial, lateral and top boundary conditions in the new "Section 4.1.2. CHIMERE set-up": "Different climatological values from the LMDZ-INCA global model [Szopa et al., 2008] or from a MACC reanalysis are used to prescribe concentrations at the lateral and top boundaries and the initial atmospheric composition in the domain."**

-It should be illustrated to what degree both are changed during the inversion. Further, a comparison or sensitivity test should be shown on what the impact of emission optimization is compared to a joint optimization with initial and boundary conditions.
**We choose not to perform such a sensitivity test. We have added text in "Section 4.2.2. Covariance matrices B and R" to explain this choice: "Based on the sensitivity test in Figure 5, the errors assigned to the CO and $NO_2$ lateral boundary conditions and to their initial conditions are set at 15%. As these errors are significantly lower than those of the emissions and as CO surface concentrations are mainly due to emissions (Figure 4), we assume a small relative influence of the correction of initial and boundary conditions on our results."**

-Although the two test cases show a reduction of the difference between the assimilated observation and the analysis, this is not a proof of the successful operation of the data assimilation algorithm. A comparison with independent observations and a table with quality measures (e. g. bias, root mean square error, cost reduction) is necessary.
**Indeed, but the main purpose of a GMD paper is less to present specific inversion studies than to describe the general modeling framework. We chose not to present evaluation for our illustration, as this is not the scope of this paper. Nevertheless, we have added sentences about the reduced mean bias between the observations and the simulation using the posterior emissions instead of the prior ones in Section 4.2.3 and in Section 4.2.4 to show the successful operation of the inversion system.**

- It is advised to perform the analysis on a few consecutive days to assess the stability and quality of the inversion on different days.
**We agree, to assess the stability and quality of the inversion, we now present a period of 7 days for the CO inversion.**

- in the description of the test cases both, initial values and boundary conditions are included in the control vector, thus, the analysis is not complete without showing these two variables. A discussion is needed about the correction for all three variables, i.e. emissions, initial values, and boundary conditions, their relative influence on the analysis and about potential limitations of the inversion.
**As already explained above, we choose not to perform such a sensitivity test. We have added text in "Section 4.2.2. Covariance matrices B and R" to explain this choice: "Based on the sensitivity test in Figure 5, the errors assigned to the CO and $NO_2$ lateral boundary conditions and to their initial conditions are set at 15%. As these errors are significantly lower than those of the emissions and as CO surface concentrations are mainly due to emissions (Figure 4), we assume a small relative influence of the correction of initial and boundary conditions on our results."**

-If the full adjoint of the chemical processes is used there should be an adjoint signal for other species than CO and NO2 as well. This must be clarified. Are these signals simply not considered or not discussed? What is the reason for not optimizing NO emissions then?

**The adjoint compute the sensitivity to all the components of the x vector. In the inversion we made for the former version of this paper, we chose to only infer NO$_2$ emissions. As both the reviewers have been disturbed by this illustration, CO, NO and NO$_2$ are now included in the x vector and we now present inversion for NO$_x$ emissions. Other species could be considered for other studies.**

-the model resolution of 0.5 x 0.5 square degrees seems to be a bit coarse for anthropogenic emission assessments. Is nesting available? A discussion on this point is needed.

**Yes, nesting is available in CHIMERE. A number of studies for anthropogenic emission assessments have been done at even coarser resolution than 0.50x0.5°: for example, Miyazaki et al. [2017] using an approximately 2.8∘×2.8∘ resolution with the global CTM MIROC-Chem or Wang et al. [2019] using a 2.5° or 2.5° resolution with Geos-Chem.**

**We discussed about finer resolutions in the perspectives of the study.**

**Miyazaki, K., Eskes, H., Sudo, K., Boersma, K. F., Bowman, K., and Kanaya, Y.: Decadal changes in global surface NO$_x$ emissions from multi-constituent satellite data assimilation, Atmos. Chem. Phys., 17, 807–837, https://doi.org/10.5194/acp-17-807-2017, 2017.**

**Wang, Y., Wang, J., Xu, X., Henze, D. K., and Qu, Z.: Inverse modeling of SO$_2$ and NO$_x$ emissions over China using multi-sensor satellite data: 1. formulation and sensitivity analysis, Atmos. Chem. Phys. Discuss., https://doi.org/10.5194/acp-2019-879, in review, 2019.**

-a better description of the B-matrix is needed in section 4.1.3.

**We agree, we have added information about the B-matrix, in Section 3.3: " Different simple but efficient ways of building the error covariance matrix B are implemented in PYVAR-CHIMERE. The variances and correlations are defined independently. The variances are specified by the user through standard deviation coefficient (Table 1), which can be a fixed value ("fx") or a percentage ("pc") to define the diagonal standard deviation matrix $\sum$. For correction types "mult" and "scale", as well as for correction type "add" with a fixed value, the value is directly used as the standard deviation of the uncertainty in the corresponding components of the control vector. For correction type "add" with a percentage provided, maps of standard deviation of uncertainty are built by applying this percentage to the matching input fields (fluxes, initial conditions, boundary conditions). The user may also provide a script to build personalized maps of variances.**

**Potential correlations between uncertainties in different types of control variables, e.g. between fluxes and boundary conditions, and correlations between uncertainties in different species, e.g. between fluxes of CO and NO$_x$, are not coded yet. Only correlations for a given type of control variable and a given species are so far taken into account so that the B matrix is block diagonal. For a given type of control variable and a given species (in the illustration in section 4.2.2: CO, NO or NO$_2$ fluxes), spatial and temporal correlations can be defined using correlation lengths through time Lt and space Ls. Those lengths are used to model temporal and/or spatial auto-correlations using an exponentially decaying function: the correlation r between parameters and at a given location but separated by duration $d(x_i, x_j)$, or at a given time but distant by $d(x_i, x_j)$ is given by $r(x_i, x_j) = exp\left(\frac{-d(x_i,x_j)}{L}\right)$ where $L$ ($= L_T$ or $L_S$) is the corresponding correlation length. There is no correlation between uncertainties in land and**

ocean flux. . Note that the spatial correlations are computed for each vertical level independently when dealing with control variables with vertical resolution (3D fields of fluxes when accounting for emission injection heights, or boundary/initial conditions). Vertical correlations in the uncertainties in such variables have not been coded yet. Apart from this, the system assumes that temporal correlations and spatial correlations depend on the time lag and distance but not on the specific time and location of the corresponding parameters. It also assumes that the correlation between uncertainties at different locations and different time can be derived from the product of the corresponding autocorrelation in time and space.

Each block of B can thus be decomposed based on Kronecker products: $B=\sum Ct \otimes C_s \sum$ where $\otimes$ is the Kronecker product, $C_t$ and $C_s$ are the temporal and spatial correlations, respectively. The calculations involving $B^{1/2}$ are simplified in PYVAR-CHIMERE using the Eigen-decomposition of $Ct$ and $Cs$. Its square root can be calculated according to: $C_t^{1/2}= V_{Ct}D_{Ct}^{1/2}V_{Ct}^T$ (and similarly for $C_s$) (Eq 4) where $V_{Ct}$ is the matrix with the Eigenvectors as columns, and $D_{Ct}$ is the diagonal matrix of Eigenvalues of $C_t$. It is possible to chose a threshold under which the eigenvalues are truncated when computing the spatial correlations in order to save computation and memory, but not when computing the temporal correlations."

-What about correlations for initial and boundary conditions?
**There is no correlation for initial and boundary conditions in our inversions.**

Further minor comments:
- line 30: (VOCs) instead of "(VOCs))"
**It has been changed.**

-line 39: reference for (LRTAP) would be appreciated
**We added the UNECE website as a reference for LRTAP.**

-line 43: no commas
**The commas have been removed.**

- line 85: CO and NOx (instead of "CO, NOx")
**It has been changed.**

-line 88: citation van der A. [2008] is not appropriate (van der A et al. [2008]), also in the reminder of the manuscript
**It has been changed.**

-line 93/94: ".. for which variational methods are more suitable than KFs by design": a reference would be appreciated for this statement.
**This sentence has been removed**.

-line 122/123: of the current inversion (instead "of the inversion")
**It has been changed.**

-line 163: quasi-Newton (instead "quasi-Newtonian")
**It has been changed.**

-line 165: Reference for incremental 4D-var approach is appreciated
**We would like to emphasize that the PYVAR-CHIMERE system for inversion is not a 4D-VAR one. We do not need reference for incremental 4D-var approach.**

- line 203: It would be appreciated if the manuscript contains a table with the available (and adjoint) processes of CHIMERE

**The available processes of CHIMERE are already listed in Figure 1 (emissions, transport, chemistry and deposition).**

- line 227/228: better: "PYVAR, CHIMERE, and text sources are displayed in blue, orange, and grey boxes, respectively."

**It has been changed.**

- caption of figure 4: better: "Simplified scheme of how PYVAR scripts prepare the observations y using satellite data. PYVAR and text sources are displayed in blue and grey boxes, respectively."

**It has been changed.**

- line 264: Equation "Cm =" is not a correct mathematical formulation, Cm(o) is a column, xa is the state vector (a profile in this context).

**Here, Cm(0) is not a column but the vertical distribution in partial subcolumns from a chemistry-transport model at the same satellite pressure levels. xa is not the state vector but the a priori profile provided together with the averaging kernels when relevant. We kept this formulation.**

- line 290: ... days for CO and NO$_2$, respectively (instead of "... days, respectively for CO and NO2")

**It has been changed.**

- line 302: Table 1 is not control vector specific. This sentence can be removed

**Indeed, we move this sentence in Section 4.**

- line 304: for one day (instead "at a 1-day"); resolution (instead "resolutions")

**It has been changed.**

- line 313/314: a spin-up for the initial values is needed for an appropriate analysis, otherwise the model maybe to far off the observations for a suitable correction.

**We agree, we indeed performed runs with a spin-up of 10 days. We have added this information in "Section 4.1.2. CHIMERE set-up": "In order to ensure realistic fields of simulated CO and NO$_2$ concentrations from the beginning of the inversion period, runs have been preceded with a 10-day spin-up."**

- line 317: Reference for MOPITT is missing

**Indeed, we have added a reference.**

- line 328: MOPITT instead of "OMI"

**It has been changed.**

- page 12, line 4: flown instead of "flying"

**It has been changed.**

- line 368: parts (instead of "part"); present (instead of "presents")

**It has been changed.**

- page 14, last line: particularly over the Po Valley (instead ", and particularly over PoValley")

**It has been changed.**

- caption Fig. 7 d: is it really the difference between prior and posterior? Inconsistency with text (see next point)
- line 374: Fig. 5c seems to be wrong here. Is it Fig. 7d?
**Indeed, this has been corrected.**

- line 380/381: Using the full adjoint of CHIMERE, this must already be available. Please check for adjoint NO signals
**As already explained above, we now present inversion of NOx emissions.**

- line 399: remove "for example"
**It has been removed.**

---

## Author Comment (AC3) · 29 Apr 2020

Short comment #1

**We wish to thank the editor for her remark. Our answer is in bold hereafter.**

Dear authors, in my role as Executive editor of GMD, I would like to bring to your attention our Editorial version 1.2:
https://www.geosci-model-dev.net/12/2215/2019/
This highlights some requirements of papers published in GMD, which is also available on the GMD website in the 'Manuscript Types' section:
http://www.geoscientific-model-development.net/submission/manuscript_types.htmlC1

In particular, please note that for your paper, the following requirements have not been met in the Discussions paper:

• The main paper must give the model name and version number (or other unique identifier) in the title. Therefore please provide the version number of the PYVAR-CHIMERE which is actually used for the current publication in the title in your revised submission to GMD.

Yours, Astrid Kerkweg

**We now provide a version number of PYVAR-CHIMERE in the title: "Variational regional inverse modeling of reactive species emissions with PYVAR-CHIMERE-v2019".**

---

## Author Response (AR2)

[revised manuscript text omitted]

**Report #1**

I thank the authors for the numerous additions and improvements to their paper. The description of the model and inversion system is now much more complete. Nevertheless, I still see some important issues (see below) that should be addressed before I can recommend the paper for publication in GMD.

**We wish to thank the referee for his/her helpful comments and for this positive acknowledgment of the improvements of our manuscript. The full review is copied hereafter and our responses are inserted in bold.**

Main comments

1) There is an unusually large number of typos, grammatical and orthographical errors. This is quite distracting while reviewing a paper. I encourage the authors to put more efforts into this aspect in the future.

**We apologize for the inconvenience. We have conducted a much more cautious and proofreading of the new version of the manuscript to avoid it.**

2) The performance of the inversions is relatively poor, especially in the NOx case. Two possible explanations should be explored: 1) the number of iterations -- I doubt that the criterion of 90% reduction of the gradient of J is sufficient. Is computing time really such a hard constraint that more iterations cannot be tested? I would be curious to see the results for a 99% reduction.

**We now present results with reduction of the norm of the gradient of $J$ for the NO$_x$ case higher than 90%, going up to 99% (see the new Table 3 describing sensitivity tests for the NO$_x$ inversion).**

**Nevertheless, it should be noted that in some cases, the minimizer finds a local minimum. Once in such a local minimum, the minimizer runs many simulations but cannot get out into a new direction and perform more iterations. This would of course not be the case for a linear problem. We have added details in Section 2: "As shown in Figure 1, the minimization algorithm repeats the forward-adjoint cycle to get an estimate close to the optimal solution of the inversion problem for the control parameters. This approximation of the optimal estimate is found by satisfying the convergence criteria of the minimizer with a given reduction of the norm of the gradient of $J$. Nevertheless, due to the non-linearity of the problem, the minimization may reach a local minimum only, instead of the global minimum."**

2) the choice of a priori errors is perplexing: 30% in the case of NOx, 100% for CO. There is no possible justification for this. Furthermore, correlations are used for one compound, not for the other

**Our examples had been chosen to illustrate the parameters of the system configuration and the way the code works. The aim was not to provide a fully comprehensive scientific study on NO$_x$ and CO inversions. Still, our choice of prior error parameters now follows insights from previous studies. In particular, Souri et al. [2020] performed NO$_x$ inversions with prior error standard deviations set at 50% of the prior estimate of the anthropogenic emissions. For the inversion of CO emissions, the error standard deviations assigned to the prior CO emissions are set at 100%. This value of 100% has already been chosen in Fortems-Cheiney et al. [2011] and in Fortems-Cheiney et al. [2012].**

**We have added such justifications for our choices of prior errors in Section 4.2.2. Furthermore, we now present some tests of sensitivity of the NO$_x$ inversions to the prior error assigned to prior emissions and initial conditions.**

Souri, A. H., Nowlan, C. R., González Abad, G., Zhu, L., Blake, D. R., Fried, A., Weinheimer, A. J., Wisthaler, A., Woo, J.-H., Zhang, Q., Chan Miller, C. E., Liu, X., and Chance, K.: An inversion of NO$_x$ and non-methane volatile organic compound (NMVOC) emissions using satellite observations during the KORUS-AQ campaign and implications for surface ozone over East Asia, Atmos. Chem. Phys., 20, 9837–9854, https://doi.org/10.5194/acp-20-9837-2020, 2020.

Fortems-Cheiney, A., Chevallier, F., Pison, I., Bousquet, P., Saunois, M., Szopa, S., Cressot, C., Kurosu, T. P., Chance, K., and Fried, A.: The formaldehyde budget as seen by a global-scale multi-constraint and multi-species inversion system, Atmos. Chem. Phys., 12, 6699–6721, https://doi.org/10.5194/acp-12-6699-2012, 2012.

The small NOx emission errors likely explain the poor match of a posteriori NO2 with OMI. I think we can all agree that the choice of a priori errors and correlations is arbitrary to some (large) degree. This is why it is important to assess whether the inversion results are dependent on such choices. Clearly this exploration is not considered important here, which I think is a mistake. Especially in view of the weak bias reduction achieved with the current setup.

**We now present tests of sensitivity to the amplitude of prior errors assigned to prior initial conditions and emissions to provide insights the impact of these choices on the**

**inversions results. We also performed a test of sensitivity to spatial correlations in these prior errors.**

**However, the biases between OMI and simulated NO₂ tropospheric columns is a complex topic that is not related to our CHIMERE simulations only [Huijnen et al., 2010; Souri et al., 2020; Elguindi et al., 2020] and that would require very careful analysis. Addressing it properly is thus clearly out of the scope of this paper.**

**We have added text: "The biases between OMI and simulated NO₂ tropospheric columns is a complex topic that is not related to our CHIMERE simulations only [Huijnen et al., 2010; Souri et al., 2020; Elguindi et al., 2020]. It requires a fully comprehensive scientific study which is out of the scope of this paper. »**

Huijnen, V., Eskes, H. J., Poupkou, A., Elbern, H., Boersma, K. F., Foret, G., Sofiev, M., Valdebenito, A., Flemming, J., Stein, O., Gross, A., Robertson, L., D'Isidoro, M., Kioutsioukis, I., Friese, E., Amstrup, B., Bergstrom, R., Strunk, A., Vira, J., Zyryanov, D., Maurizi, A., Melas, D., Peuch, V.-H., and Zerefos, C.: Comparison of OMI NO₂ tropospheric columns with an ensemble of global and European regional air quality models, Atmos. Chem. Phys., 10, 3273–3296, https://doi.org/10.5194/acp-10-3273-2010, 2010.

Elguindi, N., Granier, C., Stavrakou, T., Darras, S., Bauwens, M., Cao, H., et al.: Intercomparison of magnitudes and trends in anthropogenic surfaceemissions from bottom-up inventories, top-down estimates, and emissionscenarios. Earth's Future, 8,e2020EF001520. https://doi.org/10.1029/2020EF001520, 2020.

Minor comments

Figures 4-5 Is it a coincidence that the right panel of Fig. 4 presents high values off the coast of Egypt, which is where the MACC-based run is most different from the standard one using LMDZ-INCA? It looks like the run without emissions used boundary and initial conditions from MACC.

**Simulations with null emissions have been conducted with two different sets of boundary conditions LMDz-INCA and MACC. In Figure 4, indeed, the one that is shown used boundary and initial conditions from MACC. It has been corrected.**

l 365 Why is the median appropriate to take proper account of the AK ?

**For each model grid cell, we have several observations and corresponding averaging kernels. Taking the average of both would not make sense, as the averaged averaging kernel would not be consistent with the average observation. Conversely, taking the median of the observations allows selecting a specific observation with its corresponding averaging kernel which is consistent.**

Technical/language corrections

The appearance of mathematical symbols appears variable throughout the manuscript. For example, the B1/2 notation of l 170 (also 173) and 293. Please use consistent fonts throughout the text and equations.

**We apologize for the inconvenience. Efforts have been put into this aspect.**

l 95 "Van" -->"van"

**It has been corrected.**

l 162 "high-non linearity" ->"high non-linearity"

**It has been corrected.**

l 162-165 : Difficult to read due to sentence within parentheses; please rephrase.

**We have removed the sentence within parentheses.**

l 200: delete "a first time"

**It has been done.**

l 204 Problem with reference (parentheses)

**It has been corrected.**

l 210 A space is missing before "The adjoint"

**It has been corrected.**

l 212 "Then, it has been parallelized. This work" -->"The code has been parallelized. This task"

**It has been corrected.**

l 213 "whole code, associatedwith" -->"entire code, associated with"

**It has been corrected.**

l 213 missing space between sentences

**It has been corrected.**

l 214 missing space between sentence

**It has been corrected.**

l 216 "lead" -->"conducted"

**It has been corrected.**

l 219, 221, 222 Insert space after the bullet

**It has been done.**

l 219 "For the geometry" -->"Regarding geometry". Id. for next bullets

**It has been corrected.**

l 233 "infer" ->"infers"

**It has been corrected.**

Figure 2 The blue could be a bit lighter for lisibility (also in Figure 3)

**It has been done, for Figure 1 and Figure 2. Figure 3 has been removed.**

l 248 "to constrain could be" -->"to be constrained might be"

**It has been corrected.**

l 252 Insert a comma after "add"

**It has been done.**

l 256-260 I cannot understand what is explained here. What is meant by "activity maps and/or masks for regions"? What is meant by "control of budgets"? Etc. Please rephrase.

**We have rephrased the sentences: "For type "scale", the control variables are scaling factors applied to maps different from the maps of emissions used as prior input of the forward model: for example, activity maps can be used and scaled to get emissions; the obtained values are then added to the corresponding components of the model inputs. With these various types, it is possible to define the control variables as the budgets of**

**emissions for different regions, types of activities, and/or processes, which can thus be directly rescaled by the inversions, similarly to what is done in systems where the control vector is not gridded [Wang et al., 2018]).”**

l 264-265 "to define the diagonal standard deviation matrix SIGMA" : do you mean that the matrix is diagonal, or are your referring to the diagonal of the matrix? SIGMA has not been defined previously.

**We do not agree, SIGMA was defined l256-258 with the sentence "The variances are specified by the user through standard deviation coefficient (Table 1), which can be a fixed value ("fx") or a percentage ("pc") to define the diagonal standard deviation matrix $\sum$."**

l 281 Insert space before "where"

**It has been done.**

l 291 Subscript "t" in "Ct"

**It has been corrected.**

l 293 "The calculations involving B1/2" : it would be helpful to use equation numbers, to help the reader figure out where such calculations were mentioned.

**It has been done.**

l 294 Subscripts for Ct and Cs

**It has been corrected.**

l 298 "computation time"

**It has been corrected.**

l 313 "Simplified scheme describing how...". "prepares" -->"prepare".

**It has been corrected.**

l 313 In the colored box, what does mean "PYVAR y building"?

**Figure 3 has been removed.**

l 344 Insert a space before "OMI"

**It has been corrected.**

l 344 "to present an illustration"

**It has been corrected.**

l 381 and 383 "come from" --> e.g. "are obtained from"

**It has been corrected.**

l 388 Insert a space after "site"

**It has been corrected.**

l 391 "strongly driven" -->"strongly influenced"

**It has been corrected.**

Figure 4, the right panel is denoted using "a)" instead of "b)", please correct.

**It has been corrected.**

Fig 4 & 5 The legend could be more clear, e.g. "CO surface concentrations between 1-7 March 2015, simulated by CHIMERE...". You might drop "over Europe"

**It has been corrected. The new legend for Figure 4 is: "Mean CO surface concentrations from the 1$^{st}$ to the 7$^{th}$, March 2015 simulated by CHIMERE a) with anthropogenic and biogenic emissions, and b) without emissions, in ppbv, at the 0.5°x0.5° grid-cell resolution." The new legend for Figure 5 is: "Mean CO surface concentrations averaged from the 1$^{st}$ to the 7$^{th}$, March 2015 simulated by CHIMERE using for initial and boundary conditions, a) the climatological values from the LMDZ-INCA global model b) the climatological values from a MACC reanalysis, in ppbv, and c) the relative differences between these two simulations , in %, at the 0.5°x0.5° grid-cell resolution ».**

Figure 5 The legend says "relative difference" but the figure shows "ppbv"... Which one is it? In any case, the color scale is poorly chosen since almost all values are positive. Relative differences of 15% are still significant.

**The legend has been corrected. The color scale has also been changed.**

**The sentence describing Figure 5 has been changed: "the results were similar, with maximum relative differences in concentrations of about 15% over continental land (Figure 5c)."**

l 406 Drop "of their equivalents"

**It has been done.**

l 408-411 What is meant by "general trend in emissions"? Furthermore, if the underestimation persists throughout the year, it might still be due to specific activity sectors, isn't it? Please rephrase in a more logical manner.

**The paragraph has been changed: "These discrepancies might be due to different causes, which can all interact. A source of uncertainties is related to the observations. For example, satellite data inter-comparison studies reveal large differences between different retrievals of the same compound [Qu et al., 2020]. It can be explained by uncertainties from the CTM (e.g., through the underestimation of the atmospheric production or the underestimation of the species lifetime). It could also be explained by an underestimation of the anthropogenic emissions in the BU inventory."**

Qu, Z., Henze, D. K., Cooper, O. R., and Neu, J. L.: Improving $NO_2$ and ozone simulations through global $NO_x$ emission inversions, Atmos. Chem. Phys. Discuss., https://doi.org/10.5194/acp-2020-307, in review, 2020.

l. 412 "chemistry with OH" rephrase.

**A shown just above, the paragraph has been changed.**

Figures 6 & 7 Legend: "Mean bias" -->"Mean biases"

**It has been corrected.**

l 419 & 423 "for 7-day" is ambiguous. Only one value for the period, or several?

**We agree. We have one corrective value per pixel for the 7-day period. We have changed the description of the control vector with for example: "the CO anthropogenic emissions at a 7-day temporal resolution, a 0.5° ×0.5° (longitude, latitude) horizontal resolution, and 8 vertical levels, i.e. 101×85×8 components in x».**

l 427 "for 1-day" : as above.

**It has been corrected.**

l 433 "ofthe"

**It has been corrected.**

l 433 "We hardly have sources of estimates" : weird, re-phrase.

**To our knowledge, there are only few available studies dealing with the estimates of the uncertainties in gridded bottom-up emission inventories at the 0.5°x0.5° resolution or higher. For NO$_x$, to our knowledge, there is no study dealing with such uncertainties. For CO, Super et al. [2020] estimated uncertainties for specific grid cells at a 1km$^2$ resolution over Western Europe. This study was already cited in the text.**

**We have rephrased: "To our knowledge, there are few available studies dealing with the estimates of the uncertainties in gridded bottom-up emission inventories at the 0.5°x0.5° resolution or higher".**

l 437 "is" -->"are"

**It has been corrected.**

l 439 remove the commas before the year in the references.
**It has been done.**

l 449 "obtained" -->"obtain"
**It has been corrected.**

l 451 "even 100% of uncertainty lead to" -->"even an uncertainty of 100% leads to"
**It has been corrected.**

l 463 "deriven by" -->"driven by"

**It has been corrected.**

l 480 Space missing after Portugal
**It has been corrected.**

l 500 Please write 1.4Ee+15 as 1.4(times)10^{15}. Same elsewhere in the paper (e.g. legend of Figure 7)

**It has been corrected.**

**Report #2**

General Comments:

This manuscript presents an added inverse modeling system capability for CHIMERE based on PYVAR, which has been successfully used with LMDz for GHG flux estimation. The manuscript describes its model development and show some very promising results for CO and NOx. This version of the manuscript has improved relative to original submission, addressing several of previous reviewers comments. I commend the authors for this revision. However, while this is relevant to the scope of GMD, I have several concerns with regards to the clarity and presentation of its description and substantive presentation of results, especially on quantifying some uncertainties or describing its fidelity. I suggest minor revisions to address the following concerns before publication:

**We wish to thank the referee for his/her helpful comments and for this positive acknowledgment of the improvements of our manuscript. The full review is copied hereafter and our responses are inserted in bold.**

1) Description of the system could be improved by:
a) clear differentiation of new developments in PYVAR for reactive species. In its current form, it appears to be just an implementation of PYVAR to CHIMERE. If so, detailed testing strategies is warranted to show the fidelity of i) CHIMERE adjoint/TLM, ii) PYVAR global minimization especially for the current application to reactive species. How are these tested and quantified?

**The "PYVAR" framework in PYVAR-CHIMERE is actually quite different from that of the PYVAR system that has been used along with LMDZ. The development of an inversion system based on the PYVAR code and on CHIMERE that was suitable for our regional applications required new options for the control and observation vectors, for the set-up of uncertainties and for the interfaces with the model (that are detailed in this paper). This explains that the development of PYVAR-CHIMERE did not just consist in plugging CHIMERE to the PYVAR code.**

**Detailed testing strategies for the TL and for the adjoint are now described in Section 3.3.**

b) clear differentiation between PYVAR and 4D-Var. How is PYVAR different than other 4DVar approaches? What are its advantages and disadvantages? What are its strengths and limitations? Please add some context.

**The mathematical scheme we use to control the concentration initial and boundary conditions along with the surface fluxes using the chemistry transport model and its tangent and adjoint codes is very similar to the 4D-Var schemes used in meteorological**

data assimilation. However, for GHG/pollutant surface flux variational inversions, the main part of the control vector is fluxes over the whole data assimilation period and not the initial conditions of the state vector of the dynamical concentrations. The dynamical model connects these controlled fluxes to the observation variables (concentrations over the whole data assimilation period). Therefore, the term "4D-Var" does not really apply. For this reason, the atmospheric inverse modeling community rather uses the term "variational inversion". We thus assume that the question here is about the differences between PYVAR and other variational inversion systems.

Historically, inversion systems have been built around specific CTMs and to our knowledge, there is no study comparing different variational assimilation codes around the same CTM. In general, variational codes differ mostly in the options and features for the definition of the control and observation vectors, or of the prior and observation errors. The use of different CTMs can bring different constraints for such parameters. The underlying minimization algorithms are generally identical (CONGRAD or M1QN3 in most cases, with minor differences in implementation). With CMAQ [Hakami et al., 2007], PYVAR-CHIMERE is one of the few systems based on an Eulerian area-limited CTM. The main strengths of PYVAR-CHIMERE come from the strengths of CHIMERE and from its high modularity for the definition of the control vector (as detailed in this paper). CHIMERE is an extremely flexible code, in particular for the definition of the chemical scheme; chemical reactions are defined in a configuration file and can be switched on/off, added/removed smoothly, making it especially relevant for air quality studies.

We have added information about CHIMERE in the introduction: "CHIMERE is dedicated to the study of regional atmospheric pollution events [Ciarelli et al., 2019, Menut et al., 2020], included in the operational ensemble of the Copernicus Atmosphere Monitoring Service (CAMS) regional services. The main strengths of PYVAR-CHIMERE come from the strengths of CHIMERE and from its high modularity for the definition of the control vector. CHIMERE is indeed an extremely flexible code, in particular for the definition of the chemical scheme."

c) Clear description of Figure 3. In its current form, it shows a list or table of variables and parameters. It would be clearer if details on this Figure are discussed. What do these variables represent? Are there utilities in PYVAR system that are used? If so, it may be better to name and describe those utilities.

**Figure 3 has been removed, as the Section 3.4 already well describes the calculations of the equivalents of the observations.**

d) clear mathematical expressions and their representation (including consistent use of nomenclature). In particular, how does the time component of the cost function (i.e., 4DVAr versus PYVAR) treated in PYVAR? Can the "add", "mult", and "scale" be incorporated in the cost function notation? Where are these correction estimated and applied in the algorithm?

**See our point regarding the use of the term "variational inversion" rather than "4DVAr".**

**The control vector and observation vectors gather fluxes and concentrations (respectively) over the whole data assimilation temporal window. Decomposing the cost function into a sum of comparisons between the projection of the control variables in the observation space and the observations at different times, similarly to what is often done in meteorology, would actually make it quite confusing and uselessly complicated. The *H* operator and the H matrix bear the "time component". We think that the text makes it clear. We now refer to the paper of Rayner et al. [2019] lying the basis for variational atmospheric transport inversion to avoid having to redevelop such fundamental points.**

**We think that developing the details of *H* to show how different its columns can be in practice depending on the options for each type of control variable 'add', 'mult', and 'scale' (i.e. depending on the way the components of the control vector are defined) could be uselessly complicated (e.g., requiring a lot of indices, blocks of *H*, and different types of notations associated to the different types of components in x). In the case of 'add', the corresponding variables in the control vector and B matrix are in the physical space: in the case of surface emissions, molecules.cm$^{-2}$.s$^{-1}$ are stored in the control vector. For 'mult', the corresponding variables in the control vector and B matrix are unitless numbers: these numbers are multiplied by reference emissions before running CHIMERE. The 'scale' option is similar to the 'mult' option, but applies to activity maps, instead of reference emissions.**

**Figure 2 has been updated and now better represents where these corrections are applied in the algorithm.**

Rayner, P. J., Michalak, A. M., and Chevallier, F.: Fundamentals of data assimilation applied to biogeochemistry, Atmos. Chem. Phys., 19, 13911–13932, https://doi.org/10.5194/acp-19-13911-2019, 2019.

2)      Presentation of results could be improved by:

a) Some diagnostics to check optimality of the inversion algorithm. For example,
i) posterior error covariances – if calculated especially error reduction estimates,
**As already discussed in Section 2, posterior error covariances are not a straight forward product of variational inversion systems. We have added a reference to Rayner et al. [2019] in this paragraph. We have also added details: "Nevertheless, it should be noted that the cost of the Monte Carlo experiments used to derive these posterior uncertainties is huge."**

**We documented such computations and tests of consistency in Broquet et al. [2011] and Kadygrov et al. [2015]) for $CO_2$ and the assimilation of data from surface networks. However, such an exercise here for satellite data and highly reactive species is much more difficult.**

ii) RMSEs.This reviewer understands that comparison with independent measurements is beyond the scope of this paper. But at least provide some indication of its optimality. Results on scaling factors and increments can only be interpreted if compared to independent datasets and other approaches. However, one can show for example that the minimization reached its minimum or show the breakdown of the cost function to show that the observations are able to constrain some elements of the control vector. In its current form, this is shown qualitatively in Figure 5 to 8. At least quantify these by statistics other than mean biases. These are very promising results and should be highlighted more.

**We agree that the decrease of the misfit to the observation can be an indicator of the good behavior of the system. We have added diagnostics to check the efficiency of our system, such as RMSEs, standard deviations and correlations. We have added Table 3 and Table 4, respectively for CO and $NO_2$.**
**Sentences have been added in Section 4.2.3 for CO: "Over this area (see the grey box in Figure 6), the mean bias between the simulation and the observations has been reduced by about 27% when using the posterior emissions (mean bias of 11.6 ppbv, Table 3) instead of the prior emissions (mean bias of 15.9 ppbv, Table 3). The RMSE and the standard deviation have been reduced by about 50% and the spatial correlation has been strongly increased."**

Sentences have also been added in Section 4.2.4:"Over this area (see the purple box in Figure 6), the mean bias between the simulation and the observations has been reduced by about 24% when using the posterior emissions (mean bias of $1.9 \times 10^{15}$ molec.cm$^{-2}$ against $2.6 \times 10^{15}$ molec.cm$^{-2}$ with the prior emissions, Table 5). The RMSE and the standard deviation have been reduced by about 7%. The correlation has not been improved. Even with high emission increments, the impact on the tropospheric columns is consequently rather small. The posterior emissions and their uncertainties will have to be evaluated and may bring hints to the cause of the discrepancies between simulated and observed NO$_2$ tropospheric columns."

As an indication, in the illustration for NO$_x$ inversion (inversion E), J decreases from 622 to 594. Jo, the part of the cost function corresponding to the misfits to observation, is decreased by the inversion from 622 to 567. Jb, the component corresponding to the misfits to the prior estimate of the fluxes, increased from 0 to 27.

b) Spell and grammar checks, as well as clearer formats of tables.
We apologize for the inconvenience. We have conducted a much more cautious and proofreading of the new version of the manuscript to avoid it.

Specific Comments:

1) Abstract: Please add some numbers for your results (especially error reduction)
Information about the biases has been added in the abstract: "In these cases, local increments on CO emissions can reach more than +50%, with increases located mainly over Central and Eastern Europe, except in the south of Poland, and decreases located over Spain and Portugal. The illustrative cases for NO$_x$ emissions lead to large local increments (> 50%), for example over industrial areas (e.g., over the Po Valley) and over the Netherlands. The good behavior of the inversion is shown through statistics on the concentrations: the mean bias, RMSE, standard deviation and correlation between the simulated and observed concentrations. For CO, the mean bias is reduced by about 27% when using the posterior emissions, the RMSE and the standard deviation are reduced by about 50% and the correlation is strongly improved; for NO$_x$, the mean bias is reduced by about 24% , the RMSE and the standard deviation are reduced by about 7% but the correlation is not improved."

2) Line 16: "in addition to greenhouse gases". I understand that PYVAR has been used for GHG before, but I don't think it's used here. Good for the intro but perhaps not in the abstract. Focus on what exactly is new here.
**The reference to greenhouse gases has been removed.**

3)  Introduction: While discussion on emissions and inversions can be interesting, some of these are general statements which may not be exactly what this paper is specifically addressing and demonstrating. Focus on what issues/problems exactly the study addresses. For example, is this study addressing high resolution emission estimations (at 0.5 degrees?) or O3 or GHG? I understand that some of these statements motivate future work but perhaps make it more concise and focus more on what exactly does this paper at its current form address in terms of scientific problems?

**Ozone ($O_3$) is mentioned in the first lines of the introduction to describe general air quality issues. We choose to keep it. GHGs are mentioned to describe the evolution of the PYVAR-CHIMERE system, initially developed for the quantification of fluxes of long-lived GHG species such as $CO_2$ and $CH_4$. We also choose to keep it.**

**We agree that a 0.5° resolution is not a high spatial resolution. We have changed the sentence: "Here, we present the Bayesian variational atmospheric inversion system PYVAR-CHIMERE for the monitoring of anthropogenic emissions of reactive species at the regional scale".**

Why are other approaches not sufficient? What are the limitations of these approaches?
**Ensemble, reduced rank Kalman filter or analytical methods are popular in the inverse modeling community. Analytical methods and Kalman filters rely on strong assumptions of the linearity of the response of the observation operator to correction to the prior fluxes, which can be problematic for inverse modeling problems with highly reactive species. All these methods are low or reduced rank inversion techniques that rely on small control vectors or on a strong reduction of the space in which the inversion searches for its optimal solution. Variational inversion systems allow solving for high dimensional problems, typically solving for the fluxes at high spatial and temporal resolution, which can be critical to fully exploit satellite images. We have added this last sentence in the introduction.**

4) Line 102. Check spacing
**It has been done.**

5) Line 121. "in input of the CTM". Please clarify
**We have removed this part of the sentence.**

6) Line 124: "during the inversion process (surface fluxes …)" Perhaps make this two separate sentences?

**This has been done: "The control vector x contains these variables to be optimized during the inversion process. It can be surface fluxes but it may also include initial or boundary conditions for example, as explained in Section 3.3."**

7) Line 121-134: parameters versus variables? Are they the same?

**They are the same, but we have homogenized the sentences by keeping the same vocabulary "parameters".**

8) Line 130: "observation errors". Why not call it model-data mismatch?

**Observation errors is the appropriate vocabulary here.**

9) Line 132: "given their prior estimates, the observations, the CTM and the associated uncertainties". First, it may be better if "the" article is omitted on the succeeding segments of the sentence. Second, "given the CTM" may not be accurate way to describe this minimization.

**The sentence has been changed: "In statistical terms, the inversion searches for the most probable estimate of the control parameters given their prior estimates, observations, CTM and their associated uncertainties".**

10) Line 133: "in the following". Please clarify or omit.

**It has been removed.**

11) Line 135: What is xb? There is no mention of time in this section. Be consistent in notations (italic versus non-italic, bold vs not bold) for all expressions not just Eq. 1. What are the dimensions of these vectors and matrices?

**$x_b$ is now defined with the sentence: "The prior information about the parameters x to be optimized during the inversion process is given by the vector $x_b$".**

**The notations have been checked. The dimensions of these vectors and matrices depend on the inversion configuration chosen by the user. Illustrations are given in Section 4.**

12) Line 137: "state vector x". Is this the same as control vector? What do you mean by state? Is a parameter a state?

**It was a mistake. We have changed the sentence: "*H* is the non-linear observation operator that projects the control vector x onto the observation space".**

13) Line 139: "includes the CTM". Please clarify.

**The sentence has been changed: "the observation operator includes the operations performed by the CTM in linking the emissions to the concentrations and any other**

**transformation to compute the simulated equivalent of the observations such as an interpolation or an extraction and averaging of the simulated concentrations fields".**

14) Line 157: I suggest to may Eq 2 as a separate line. Should B be bold?

**Equation 2 is now presented as a separate line. B indeed should be bold, it has been corrected.**

15) Line 142: "errors are assumed to be centered and to have Gaussian distribution". What do you mean by centered? Unbiased? Is it necessary to assume Gaussianity? If so, then why not just solve the analytical solution? Because H is non-linear?

**We indeed mean unbiased. The sentence has been corrected.**

**The fact that $H$ is of high dimension mainly justifies the use of a variational approach. See our answer above.**

16) Line 159: "optimal solution". What does optimality mean here?

**The sentence has been detailed: "As shown in Figure 1, the minimization algorithm repeats the forward-adjoint cycle to get an estimate close to the optimal solution of the inversion problem for the control parameters. This approximation of the optimal estimate is found by satisfying the convergence criteria of the minimizer with a given reduction of the norm of the gradient of $J$. Nevertheless, due to the non-linearity of the problem, the minimization may reach a local minimum only, instead of the global minimum."**

17) Line 163: Careful on spacing between words

**It has been corrected.**

18) Line 172: I suggest to make the modified equation 2 as a separate line. How is this linearization implemented? i.e., where is this linearization point relative to the iteration interval? Does making shorter intervals improve minimization and representation of non-linearity?

**We do not understand this question and in particular what the reviewer calls the "iteration interval". CHIMERE uses inputs, such as emission fluxes, at an hourly resolution. Then, the computations of mixing and chemistry are done at a finer time resolution. The linearization of CHIMERE relative to anthropogenic emission fluxes for example is done for linearization points which are the emissions at the beginning of each hour and the computations for mixing and chemistry are run for the sub time steps inside each hour. The number of sub-hourly time steps is the same for the forward and**

**tangent-linear runs and is chosen to ensure a good representation of the physical and chemical processes in all cases without inducing too large a computing time.**

**The modified equation 2 is now presented as a separate line.**

19) Line 173: what do you mean by "norms".

**Here, "norm" is the mathematical term. It refers to Euclidean or $l2$ norms, as explained in Gilbert and Lemaréchal. [2009].**

Gilbert, J.C.†and Lemaréchal, C.: The module M1QN3, Version 3.3, https://who.rocq.inria.fr/Jean-Charles.Gilbert/modulopt/optimization-routines/m1qn3/m1qn3.pdf, 2009.

20) Line 175: how do you address local minimum?

**There is no way to know when you are in a local minimum rather than the global one. In practice, sensitivity studies can be performed.**

21) Line 177-184: If there's an estimate of posterior uncertainty in this system, is this used in the study? Please state which approach is used.

**Please refer to our response above.**

22) Figure 1. Are B and R fixed? Caption has bold fonts.

**B and R are matrices. They are represented with bold fonts**.

23) Line 199: "without chemistry a first time". Please clarify.

**We have removed "a first time" in this sentence.**

24) Section 3.2 How do you diagnose if these adjoints are calculated accurately? Are there tests conducted for this purpose?

**We have added a subsection describing the different tests of our system: "3.3. Accuracy of tangent-linear and adjoint codes**

**Different procedures have been implemented to test the accuracy of the tangent-linear (TL) and adjoint codes.**

**To test the linearity of the TL, we compute a Taylor diagnostic. It consists in computing the TL at $x_0$ for given increments $\Delta x$, $dHx_0$ $(\Delta x)$, then the TL at $x_0$ for $\lambda \times \Delta x$ with $\lambda$ an arbitrary small number, $dHx_0(\lambda \Delta x)$.**

**Theoretically, if the TL is well coded, $\lambda dHx_0(\Delta x)=dHx_0(\lambda \Delta x)$ by definition. In practice, the difference must be lower than 10 times the precision of the machine on which it is run.**

**The adjoint code is also tested, by verifying that $<H.\Delta x, H.\Delta x>=<\Delta x, H^T H.\Delta x>$ where $H^T$ stands for the adjoint at x. What is actually computed is the ratio of the difference between the two scalar products to the second one and the accuracy of the computation. The difference should be a few times the precision of the machine on which it is run."**

25) Line 207-216. Please check spacing between words.

**It has been corrected.**

26) Line 214: "lead with". Please clarify.

**We have changed the sentence: "Changes have been implemented in the forward CHIMERE code embedded in PYVAR-CHIMERE to match the requirements of the studies conducted with this system".**

27) Line 217-222. While important, I suggest to have them numbered but part of the paragraph rather than bulleted. And please elaborate each one.

**We have changed the sentences: "Compared to the CHIMERE 2013 version [Menut et al., 2013], the most important of these changes are, regarding geometry, the possibility of polar domains and the use of the coordinates of the corners of the cells instead of only the centers, allowing the use of irregular grids. Regarding transport, the non-uniform Van Leer transport scheme on the horizontal has been implemented, which is consistent with the use of irregular grids. Finally, various switches have been added to keep the system consistent for GHG studies. For example, we can avoid going into the chemistry, deposition or wet deposition routines when the focused species do not require them (e.g. no chemistry for methane or carbon dioxide at a regional scale)."**

28) Line 221: "when no species requires them". Please clarify. Do you mean for GHG for all – chem, dep? Or for a particular species that do not have either of these processes?

**We mean for species that do not have either of these processes. As seen above, the sentences have been changed.**

29) Line 224: "currently operational". Please clarify. Does this mean it is used in operational mode to forecast and predict? Also, is there a particular version of PYVAR and CHIMERE and PYVAR-CHIMERE used in this study?

**PYVAR-CHIMERE is not used in operational mode to forecast and predict. We have changed the sentence: "PYVAR-CHIMERE is currently implemented with a full module of gaseous chemistry".**

**The CHIMERE version has been already described: "The development and maintenance of the adjoint means that the version used is necessarily one or two versions behind the distributed CHIMERE version". The inversion code in PYVAR-CHIMERE follows the general framework of the PYVAR codes documented in the series of paper using LMDZ from Chevallier et al. [2005] to Zheng et al., [2019] (and a significant amount of lines of codes). But in practice, this code differs in various ways, in particular due to its specific adaptations to CHIMERE and the types of problems it can deal with (e.g. non-linear), and regarding the definition of the control and observation vectors, or of the B and R matrices. In some sense, the name PYVAR-CHIMERE can be taken as a version of the PYVAR code. As given in the title, we have named the PYVAR-CHIMERE version "PYVAR-CHIMRE-v2019".**

Chevallier, F., M. Fisher, P. Peylin, S. Serrar, P. Bousquet, F.-M. Bréon, A. Chédin, and P. Ciais: Inferring CO2 sources and sinks from satellite observations: method and application to TOVS data, *J. Geophys. Res*., 110, D24309, doi:10.1029/2005JD006390, 2005.

Zheng, B., Chevallier, F., Yin, Y., Ciais, P., Fortems-Cheiney, A., Deeter, M. N., Parker, R. J., Wang, Y., Worden, H. M., and Zhao, Y.: Global atmospheric carbon monoxide budget 2000–2017 inferred from multi-species atmospheric inversions, *Earth Syst. Sci. Data*, 11, 1411–1436, https://doi.org/10.5194/essd-11-1411-2019, 2019.

30) Table 1 is very informative. Please format accordingly, especially separating the header as it becomes confusing to read. Not sure if the "example of the definition.." row should be there. Can it be in the title?

**We agree, it has been changed.**

31) Section 3.3. Discussion of correction types is very informative as well. Is it possible to show how these are related to Eq. 1 to 3?

**As explained above, the 'add', 'mult', and 'scale' correction types do not need to be explicitly included in the cost function.**

Isnt it that the control vector consists of elements –corresponding to each grid point and species? If so, how is "scale" implemented to maps or masks for regions?

**We have rephrased the sentences: "For type "scale", the control variables are scaling factors applied to maps different from the maps of emissions used as prior input of the forward model: for example, activity maps can be used and scaled to get emissions; the obtained values are then added to the corresponding components of the model inputs. With these various types, it is possible to define the control variables as the budgets of emissions for different regions, types of activities, and/or processes, which can thus be directly rescaled by the inversions, similarly to what is done in systems where the control vector is not gridded [Wang et al., 2018])."**

32) Line 254: "which is similar to the control vector of budgets…" Please elaborate.

**Please see the answer above.**

33) Line 256: "adding the obtained values to the …" please rephrase.

**Please see the answer above.**

34) Line 259: "standard deviation coefficient". Please clarify. Is it really a coefficient? And since this is an error covariance matrix, should the diagonal elements be error variance not error standard deviation?

**Indeed the diagonal elements are variances, as already explained in Section 3.4: "The variances are specified by the user through standard deviation coefficient (Table 1), which can be a fixed value ("fx") or a percentage ("pc") to define the diagonal standard deviation matrix $\sum$."**

35) Line 260-262: Very important statement. But please elaborate or rephrase.

**The description of the correction types has been detailed above.**

36) Line 266: "variances". Are these error variances?

**Yes, they are.**

37) Line 270: " error correlation between fluxes of CO and NOx, are not coded yet". Please elaborate on its potential effect on your estimation?

**We can not quantify this potential effect at this stage and to our knowledge, there is no study about this in the literature.**

38) Line 296: How about calling this "Observation Operators"?

**We do not agree. The section title remains "Equivalents of the observations".**

39) Line 298: Please note spacing between words.

**It has been corrected.**

40) Section 3.4. I think this is very relevant. Please elaborate Figure 3. In its current form, it is not clear what this Figure represents and how we can use it to interpret results. I think coding of these operators is a vital step in the assimilation and should be given more emphasis. Are these utilities also available? How good are the adjoints of these operators? Are there tests to diagnose their accuracy?

**As said above, Figure 3 has been removed. We have added a subsection describing the different tests of our system: "3.3. Accuracy of tangent-linear and adjoint codes".**

41) Please check bold fonts in line 311 to 312

**It has been done.**

42) Line 314-318: Please highlight in your notations if these are scalars or vectors.

And please add corresponding dimensions. What is the difference between small ($c\_m(o)$) and big $C\_m(o$. What is $x\_a$?

**There is no difference between these two notations. Notations have been homogenized. As already noticed in Section 3.5, $x_a$ is the prior profile provided with averaging kernels, when this is relevant.**

43) Line 328-334: This is also informative. Is there a reference for parallelization approach in PYVAR and CHIMERE?

**The parallelization approach for CHIMERE is described in the Section 2.2 of Menut et al. [2013].**

How does it scale with more CPUs? 4 hours seem to be a long time, isn't it? Please elaborate and compare with other systems.

**The optimal number of CPUs for the parallelization of the transport scheme depends on the size of the tiles (for the Van Leer scheme, they must be at least 6 grid cell large because of the upwind and downwind information required) and also of the technical characteristics of the machine, because of the time required to exchange halos. A set-up with many tiles on many CPUs requiring large amounts of exchanges for halos may be less efficient than a setup with less tiles (each being larger). The performances quantified on a given type of machine is not transposable to another since they are sensitive to CPU types etc.**

44) Line 336-343. Check spacing between words.

**It has been done.**

45) Line 391-392. Why are they not different?

**The sentence has been changed: "To characterize the uncertainties in the concentration fields due to the initial and lateral boundary conditions, we performed a sensitivity test by using either climatological values from LMDZ-INCA or a MACC reanalysis: maximum relative differences in concentrations of about 15% over continental land are estimated (Figure 5c)."**

46) Figure 5 caption. "differences are in %" is in contrast to the units in the figure.

**It has been corrected.**

47) Figure 6 and 7. Is it possible to show difference plots? And more statistics (RMSEs, correlation, bias? Error reduction? Are these really surface concentrations? They are column measurements, right?

**Figure 7 indeed represents tropospheric columns, the legend has been corrected.**

What about initial conditions? Has this change as well since these are part of the control vector? Superscript on units?

**The initial conditions are slightly changed. This is now described in Section 4.2.2: "With prior error standard deviations assigned to 15% of the initial conditions, the changes in initial conditions are very small (not shown) and do not affect the posterior emissions (test B, Figure 8)."**

**The superscripts on units have been corrected.**

48) Section 4.2. Should this be presented prior to section 4.1.3 since some of the plots are for the posterior estimates?

**We do not agree, we have kept the sections as initially presented.**

49) Section 4.2.1. Can this be summarized in a table and discuss a little bit in the text as to the rational of the choice of these parameters? Am I to assume that NOx emissions are estimated only for 1 day, and all days are the same? For CO, what do you mean by 7-day? Average? How are emissions incorporated in CHIMERE in terms of time? Is there a distribution? i.e., diurnal and weekly cycle?

**The choice of the parameters has already been summarized in Table 1. $NO_x$ emissions are indeed estimated for one day in our illustration. As mentioned above, we have changed the description of the control vector in Section 4.2.1. Choices for prior error statistics are now better described in Section 4.2.2: "… different sensitivity tests described in Table 3 have been performed for the construction of the B matrix. For both**

the prior NO and NO₂ emissions at 1-day and 0.5° resolution, the prior error standard deviations are first assigned to 50% of the prior estimate of the emissions (test A), as in Souri et al. [2020]. Sensitivity tests have also been performed with prior error standard deviations assigned to 80 and 100% of the prior estimate of the emissions (test C and test D, respectively, Figure 8). With a prior error standard deviation assigned to 15% of the initial conditions, the changes in initial conditions are very small (not shown) and do not affect the posterior emissions (test B, Figure 8). As indicated in Section 3.4 and in Table 1, it is possible to use correlations in B, as in Broquet et al. [2011], in Broquet et al. [2013] and in Kadygrov et al. [2015].We have also demonstrated the strong impact of spatial correlations, defined by an e-folding length of 50km over land and over the sea, on our inversions results (test E, Figure 8)."

Description about the TNO anthropogenic emissions have been added in Section 4.1.2:"The prior anthropogenic emissions for CO and NO$_x$ emissions are obtained from the TNO-GCHco-v1 inventory [Super et al., 2020], the last update of the TNO-MACCII inventory [Kuenen et al., 2014]. This inventory is based on the EMEP/CEIP official country reporting for air pollutants done in 2017. It is an inventory at 6kmx6km horizontal resolution. From the annual and national budgets, each sector is assigned to a specific proxy to quantify the spatial variability of the emissions within each country. Temporal profiles are also provided per GNFR sector code (variations due to the month, weekday and hour). Following the Generation of European Emission Data for Episodes (GENEMIS) recommendations [Kurtenbach et al., 2001; Aumont et al., 2003], NO$_x$ emissions are speciated as 90% of NO, 9.2% of NO₂, and 0.8% of HONO. The TNO-GHGco-v1 inventory has been aggregated to the CHIMERE grid."

50) Section 4.2.2. Please check spacing of words and bold fonts.

**It has been done.**

51) Section 4.2.3. How about emission error reduction?

**The emission error reduction is not a straightforward product of the method.**

How do you ensure that these increments are "resolved by the observations".

**The method ensures it by design. Sensitivity tests will be done later over longer periods for example with constant emissions to quantify the impact of the observations in the system.**

It would be great to see error reduction plots, if posterior error covariances are calculated.

**They aren't, as explained before.**

How about initial conditions? Did this change as well?

**See the answer above.**

52) Line 508-516. What is the implication of this to overall cost and computing and optimality of minimization including error correlation of CO and NOX (and spatial correlation against superobbing) as well as increase in dimension of control vector? This also entails using this system at higher spatiotemporal resolution, right? It would be great to have a section on limitations before future implication.

**The high-resolution imaging of TROPOMI will indeed entail using PYVAR-CHIMERE at higher spatio-temporal resolutions, but for smaller domains (i.e., over countries rather than over Europe) as a compromise between resolution and the computational cost.**

---

## Author Response (AR3)

We wish to thank the referee for his/her helpful comments and for this positive acknowledgment of the improvements of our manuscript. The full review is copied hereafter and our responses are inserted in bold. We want to recall that our examples had been chosen to illustrate, for GMD, the parameters of the system configuration and the way the code works. We present a state-of-the-art system for regional inversions of reactive species emissions and we will provide fully comprehensive scientific study on  $NO_x$  and CO inversions in the near future and in dedicated journals.

The authors state that "the biases between OMI and simulated NO2 tropospheric columns are a complex topic (...) Addressing it properly is thus clearly out of the scope of this paper." This is true, but still, the inversion of emissions is expected to bring the model much closer to the observations even when the model and data are flawed. This article should show that the behavior of the inversion system is well understood, which I am not convinced.

In the NOx inversion E, the simulated  $NO_2$  columns are increased by only about 10% over northern Germany and the Netherlands (based on Table 5 and Figure 6), despite large emission increments (>20%, possibly much more). Why this lack of sensitivity?

The inadequacies of the improvements found in this study are not related to our CHIMERE simulations only [Huijnen et al., 2010; Miyazaki et al., 2017; Souri et al., 2020; Elguindi et al., 2020]. For example, Miyakazi et al., [2017] found positive increments higher than 40% over parts of Western Europe. However, they do not improve the bias between the simulations and the OMI observations (mean bias of -0.45 1015 molec.cm-2 with and without data assimilation, see their Table 2).

We have performed a test to explain this lack of sensitivity. We have simulated the NO2 tropospheric columns with biogenic emissions from MEGAN and the anthropogenic emissions from the TNO-GHGco-v2 inventory (called PRIOR in Figure 1). We have also simulated NO2 columns with anthropogenic emissions increased by a factor 3 (called PRIORx3 in Figure 1). The ratio between these two simulations shows strong nonlinearities, blurring the multiplicative effect of our increments and explaining the lack of sensitivity. By increasing NOx anthropogenic emissions, NO2 tropospheric columns can be strongly increased (Figure 1c) and can exceed the observations values for particular pixels (e.g., for 8 pixels in the purple box of our Figure 6 in the draft). NO2 tropospheric columns can also be decreased, or only slightly increased as it is seen for example over rural areas over Spain (Figure 1c). On average, it tends to increase the concentrations by a factor that is much smaller than the factor of increase in the anthropogenic emissions. However, the patterns where the posterior tropospheric columns exceed the observations or, on the opposite are decreased or only slightly increased, explain why the inversion system does not attempt at increasing further the average level of the concentration (to decrease further the general bias to the observations), even though it accounts for the impact of non-linearities in the chemistry through the use of the M1QN3 minimization algorithm.

Several studies have reported that strong non-linear relationships exist between  $NO_x$  emissions and satellite  $NO_2$  columns [Lamsal et al., 2011; Vinken et al., 2014; Li and Wang, 2019]. This reveals that a fully comprehensive scientific study is required, by

analyzing the  $NO_x$  lifetime through processes such as the  $NO_2$ +OH reactions and/or the reactive uptake of  $NO_2$  and  $N_2O_5$  by aerosols [e.g. Lin et al., 2012; Stavrakou et al., 2013].

**Figure 1.**  $NO_2$  collocated tropospheric columns left) simulated by CHIMERE using the prior TNO-GHGco-v1 emissions, middle) simulated by CHIMERE using the prior TNO-GHGco-v1 emissions increased with a factor 3 in  $10^{16}$  molec.cm-2 and right) ratio between these two fields, at the  $0.5^{\circ}x0.5^{\circ}$ grid-cell resolution, the  $19^{th}$ , February 2015.

It is stated on I. 476-482 that the discrepancies might have different causes including biases in the observations, in the emissions, and in the model. Nevertheless, the basic assumption of inverse modeling is that errors in the emissions play the dominant part. Therefore, we expect a substantial reduction of the bias after inversion, at least if the observations do not have huge uncertainties. What are the relative uncertainties in the NO2 observations used here?

We agree. We normally expect a reduction of the bias after inversion if the observations do not have huge uncertainties. As seen in Figure 2a, the bias between the simulation and the observations is indeed reduced with posterior emissions, in particular when the OMI uncertainties are the lowest, over parts of Spain, Italy and northeastern Germany (Figure 2b). Nevertheless, the bias reduction is not as substantial as expected due to high non-linearities linked to the NOx chemistry (as explained just above). Inferring NOx emissions is therefore challenging.

We have added this Figure 2 in the manuscript (as Figure 9). We have also added details in the text: "Over this area (see the purple box in Figure 6), where the OMI uncertainties are lower than 50% (Figure 9b), the mean bias between the simulation and the observations has been reduced by about 24% when using the posterior emissions (mean bias of  $1.9 \times 10^{15}$  molec.cm-2 against  $2.6 \times 10^{15}$  molec.cm-2 with the prior emissions, Table 5, Figure 9a).

---

## Author Response (AR4)

**We wish to thank the referee for his/her helpful comments and for this positive acknowledgment of the improvements of our manuscript. The full review is copied hereafter and our responses are inserted in bold.**

I thank the authors for investigating the lack of sensitivity through additional computations. Those tests are useful. Of course there is a strong non-linearity as the increased NOx emissions tend often to increase OH and decrease the lifetime of NOx. There might be other chemical effects, which would require a more detailed scientific study, as the authors state in the conclusions. Besides chemical effects, the lack of sensitivity is further enhanced by 1) the (presumably small) contribution of non-anthropogenic emissions, and 2) the contribution of emissions during the preceding days. I think this should be mentioned in the manuscript.

**We agree that the contribution of non-anthropogenic emissions could enhance the non-linearity during a large part of the year but their impact should be insignificant for our illustration case in February, when biogenic emissions are very small.**

**We have added the following sentences in section 4.2.4: "We can conclude that the strong non-linearities of the NO$_x$ chemistry mainly explain the lack of sensitivity between NO$_x$ emissions and satellite NO$_2$ columns. Besides chemical effects, the lack of sensitivity could be also partly due to the contribution of emissions during the preceding days and the assimilation window will be widened in the near future."**

I can't say I agree entirely that the inversion system fails to reproduce the patterns of the observations \*\*because\*\* of this non-linearity. It does not help, of course, in the sense that very large emission increments are needed to overcome the negative feedbacks and match the observations. And very large increments are penalized in the cost function. In your test with anthropogenic emissions multiplied by 3, the relative emission increment was uniform, whereas in the optimization, the system is free to modify the emission distribution. In a test with infinite a priori emission errors and no spatial correlation, the system would very probably do a better job. In your setup, with correlations and conservative emission error estimates, the system finds a compromise (which is perfectly reasonable). I would appreciate if the discussion could reflect the fact that the choice of errors and correlations has a likely strong impact on the results.

**We agree and we have already demonstrated the strong impact of the prior error standard deviations and of the spatial correlations in the B matrix in section 4.2.2. We have added a sentence in section 4.2.2:** "To our knowledge, there are few available studies dealing with the estimates of the uncertainties in gridded bottom-up emission inventories at the 0.5°x0.5° resolution or higher. The characterization of their statistics in the inversion configuration is consequently often based on crude assumptions from the inverse modelers. **Defining the covariance matrices B and R is not an easy task, while incorrectly specifying these matrices has a very strong impact on the results of the inversion. Especially, the relative weights of B and R, and the spatial and temporal correlations in B influence the degree of freedom and the structure for the adjustments attempted by the inversion in the optimization process.** Consequently, as an example for the NO$_x$ inversion, different sensitivity tests described in Table 3 have been performed for the construction of the **B** matrix."

---

## Author Response (AR5)

**Editor**

Comments to the Author:

There seem to be missing corrections asked by referee 2 from version 4 of the manuscript onwards:

These points seem not to be addressed neither in gmd-2019-186-author_response-version2.pdf nor in gmd-2019-186-manuscript-version[5-7].pdf

Maybe they were not included in the uploaded version? Or am I missing something? Until 33) the points are addressed in gmd-2019-186-manuscript-version5.pdf.

**There may have been some misunderstanding. We have answered the points from 33) in October 2020, but we have thought that the resulting discussions should not lead to corrections in the manuscript. In the following, we include the answers we gave in gmd-2019-186-author_response-version2.pdf in italic. Further explanations or new answers are given in bold.**

The following points seem to be still missing:

34) Line 259: "standard deviation coefficient". Please clarify. Is it really a coefficient? And since this is an error covariance matrix, should the diagonal elements be error variance not error standard deviation?

**in gmd-2019-186-author_response-version2.pdf p54:** *Indeed, the diagonal elements are variances, as already explained in Section 3.4: "The variances are specified by the user through standard deviation coefficient (Table 1), which can be a fixed value ("fx") or a percentage ("pc") to define the diagonal standard deviation matrix $\sum$."*

**We have changed the sentence: "The variances are specified by the user through the specification of the values for the corresponding standard deviation (i.e. the diagonal matrix of standard deviations $\sum$, Table 1) which can be made in terms of fixed values ("fx") or percentages ("pc")."**

35) Line 260-262: Very important statement. But please elaborate or rephrase. What is standard deviation of the uncertainty?

**The different ways of building the error covariance matrix are detailed in gmd-2019-186-author_response-version2.pdf p10-11, from line 308 to 344. The words "standard deviation of the uncertainty" have been removed and this comment does not apply anymore.We have changed the sentence: "For correction types "mult" and "scale", as well as for correction type "add" with a fixed value, the value is directly used as the uncertainty in the corresponding components of the control vector."**

36) Line 266: "variances". Are these error variances?

**in gmd-2019-186-author_response-version2.pdf p54:** *Yes, they are.*

37) Line 270: "error correlation between fluxes of CO and NOx, are not coded yet". Please elaborate on its potential effect on your estimation?

**in gmd-2019-186-author_response-version2.pdf p54:** *We cannot quantify this potential effect at this stage and to our knowledge, there is no study about this in the literature.*

When handling CO and NO$_x$ emissions from anthropogenic combustion, depending on whether the major source of uncertainty in the CO and NO$_x$ emissions is connected to the level of corresponding activity, or to the emission factors corresponding to the conversion of activity level into emission estimates, such correlations could be high or low, and even negative. Such correlations generate some transfer of information from atmospheric data for one species to the emission of the other species, high positive correlations enhancing the overall constraint on the emissions from a given set of observations. However, there is currently no strong consensus regarding the levels of correlations and it clearly depends on the specific study cases (cities, regions, countries etc.) For other types of emitting processes, one can hardly find some correlations between uncertainties in NO$_x$ and CO fluxes.

**We have added a sentence in the text: "Such correlations increase the observation constraint on the emissions in the inversion process by transferring information from one species to the other. The level (and sometimes the sign) and thus the impact on the inversion of such correlations highly depends on the study cases, and are often debated due to the lack of precise characterization of the uncertainties in inventories of anthropogenic emissions of GHG and pollutants [Super et al. 2020]."**

38) Line 296: How about calling this "Observation Operators"?

**in gmd-2019-186-author_response-version2.pdf p55:** *We do not agree. The section title remains "Equivalents of the observations".*

**We would like to recall that there is only one observation operator, which includes the CTM but also the various steps (e.g., interpolation, averaging, application of the averaging kernels) to compute the equivalents of the observations. The section title remains: 'Equivalents of the observations".**

40) Section 3.4. I think this is very relevant. Please elaborate Figure 3. In its current form, it is not clear what this Figure represents and how we can use it to interpret results. I think coding of these operators is a vital step in the assimilation and should be given more emphasis. Are these utilities also available? How good are the adjoints of these operators? Are there tests to diagnose their accuracy? The reviewer asked for clarifications on figure 3 but it was removed altogether, and further clarifications seem to be missing. Were those considered in an intermediate version?

**in gmd-2019-186-author_response-version2.pdf p44:** *Figure 3 has been removed, as the Section 3.4 already well describes the calculations of the equivalents of the observations.*

**in gmd-2019-186-author_response-version2.pdf p55:** *We have added a subsection describing the different tests of our system: "3.3. Accuracy of tangent-linear and adjoint codes".*

**The Section 3.3 is recalled here: "Different procedures have been implemented to test the accuracy of the TL and adjoint codes. To test the linearity of the TL, we compute a Taylor diagnostic. It consists in computing the TL at $x_0$ for given increments $\Delta x$, $dHx_0 (\Delta x)$, then the TL at $x_0$ for $\lambda \times \Delta x$ with $\lambda$ an arbitrary small number, $dHx_0(\lambda \Delta x)$. Theoretically, if the TL is well coded, $\lambda dHx_0(\Delta x)=dHx_0(\lambda \Delta x)$ by definition. In practice, the difference must be lower than 10 times the precision of the machine on which it is run.**

**The adjoint code is also tested, by verifying that $<H.\Delta x,H.\Delta x>=<\Delta x,H^T H.\Delta x>$ where $H^T$ stands for the adjoint at x. What is actually computed is the ratio of the difference between the two scalar products to the second one and the accuracy of the computation. The difference should be a few times the precision of the machine on which it is run."**

42) Line 314-318: Please highlight in your notations if these are scalars or vectors. And please add corresponding dimensions. What is the difference between small (c_m(o)) and big C_m(o. What is x_a?

Matrices, vectors and scalars are still indicated with the same typeface in v7.

**All the components of the following equations are matrices or vectors, described with standard mathematical notations i.e. using bold capital letters for matrices and bold lowercase letters for vectors. We have changed the notations:** "Two types of formula, depending on the satellite observations used, have been detailed in PYVAR-CHIMERE for the use of AKs: $\boldsymbol{c_m} = \boldsymbol{AK}.\boldsymbol{c_{m(o)}}$ (Eq. 8) **or** $\boldsymbol{c_m} = \boldsymbol{x_a} + \boldsymbol{AK}(\boldsymbol{c_{m(o)}} - \boldsymbol{x_a})$ (Eq. 9) where $\mathbf{c_m}$ is the modeled column, **AK contains the averaging kernels that can be provided in the form of vector (e.g., OMI product) or matrix (e.g., MOPITT product),** $\boldsymbol{x_a}$ **is the prior state vector** (provided together with the AKs when relevant) and $\mathbf{c_{m(o)}}$ is the vertical distribution of the original model partial columns interpolated to the pressure grid of the AKs."

43) Line 328-334: This is also informative. Is there a reference for parallelization approach in PYVAR and CHIMERE? How does it scale with more CPUs? 4 hours seem to be a long time, isn't it? Please elaborate and compare with other systems.

**in gmd-2019-186-author_response-version2.pdf p55:** *The parallelization approach for CHIMERE is described in the Section 2.2 of Menut et al. [2013].*

**We have added the reference in the text: "As described in Menut et al. [2013] for CHIMERE, the model parallelization results from a Cartesian division of the main geographical domain into several sub-domains, each one being processed by a worker process."**

**in gmd-2019-186-author_response-version2.pdf p55:** *The optimal number of CPUs for the parallelization of the transport scheme depends on the size of the tiles (for the Van Leer scheme, they must be at least 6 grid cell large because of the upwind and downwind information required) and also of the technical characteristics of the machine, because of the time required to exchange halos. A set-up with many tiles on many CPUs requiring large amounts of exchanges for halos may be less efficient than a setup with less tiles (each being larger). The performances quantified on a given type of machine is not transposable to another since they are sensitive to CPU types etc.*

**We have added this sentence in the text:** *"***The optimal number of CPUs for the parallelization of the transport scheme depends on the size of the tiles and also of the technical characteristics of the machine, because of the time required to exchange halos."**

47) Figure 6 and 7. Is it possible to show difference plots? And more statistics (RMSEs, correlation, bias? Error reduction? Are these really surface concentrations? They are column measurements, right? What about initial conditions? Has this change as well since these are part of the control vector? Superscript on units?

**in gmd-2019-186-author_response-version2.pdf p56:** *Figure 7 indeed represents tropospheric columns, the legend has been corrected.*

**in gmd-2019-186-author_response-version2.pdf p58:** *The initial conditions are slightly changed. This is now described in Section 4.2.2: "With prior error standard deviations assigned to 15% of the initial conditions, the changes in initial conditions are very small (not shown) and do not affect the posterior emissions (test B, Figure 8)."*

I agree with the reviewer the it is difficult to tell the differences apart. Could at least a suitable colour map be used in order to visually appreciate the differences?

**We thought that the added Table 4 and Table 5 with statistics for the comparison were sufficient to understand the differences between simulations and observations. We have added differences maps in Figure 5 and Figure 6.**

[Figure]

*Figure 5. Mean CO collocated surface concentrations from the 1ˢᵗ to the 7ᵗʰ, March 2015 a) simulated by CHIMERE using the prior TNO-GHGco-v1 emissions and the climatological values from the LMDZ-INCA global model for initial and boundary conditions, b) observed by MOPITTv8-NIR-TIR and c) simulated by CHIMERE using the posterior emissions, in ppbv, at the 0.5°x0.5° grid-cell resolution. **Relative differences between MOPITT and d) the prior CHIMERE simulation or e) the posterior CHIMERE simulation, in %.** Statistics for the comparison between simulations and observations are given in Table 4 for the area in the purple box.*

[Figure]

*Figure 6. NO₂ collocated tropospheric columns a) simulated by CHIMERE using the prior TNO-GHGco-v1 emissions and the climatological values from the LMDZ-INCA global model for initial and boundary conditions, b) observed by OMI and c) simulated by CHIMERE using the posterior emissions, in $10^{16}$ molec.cm$^{-2}$, at the 0.5°x0.5° grid-cell resolution, the 19ᵗʰ, February 2015.* **Relative differences between OMI and d) the prior CHIMERE simulation or e) the posterior CHIMERE simulation, in %.** *Statistics for the comparison between simulations and observations are given in Table 5 for the area in the purple box.*

48) Section 4.2. Should this be presented prior to section 4.1.3 since some of the plots are for the posterior estimates?

**in gmd-2019-186-author_response-version2.pdf p56:** *We do not agree, we have kept the sections as initially presented.*

**The section 4.1.3 is about the CO Sensitivity to emissions and to initial and boundary conditions. We still prefer to keep this section before the Section 4.2 about the inversions.**

49) Section 4.2.1. Can this be summarized in a table and discuss a little bit in the text as to the rational of the choice of these parameters?

**The Table 1 summarizes our examples for the definition of the control vector and for the construction of the B matrix since the first review and Table 1 is still present in the last version of the manuscriptgmd-2019-186-manuscript-version7.pdf.**

**Section 4.2.1 and Table 1 describe the spatial and temporal resolution of our control vector. We have further improved the definition of our control vector x: "For the CO inversion, the control vector x is:**

- **the CO anthropogenic emissions at a 7-day temporal resolution, a 0.5° ×0.5° (longitude, latitude) horizontal resolution, and over the first 8 vertical levels, i.e. for each of the corresponding 101×85×8 grid cells,**
- **the CO lateral and top boundary conditions at a 7-day temporal resolution, at a 0.5° ×0.5° (longitude, latitude) resolution and over the 17 vertical levels of CHIMERE, i.e. (2x101 + 2x85) x 17 grid cells,**
- **the CO 3D initial conditions for the 1$^{st}$ March 2015 at 0:00 UTC , at a 0.5° ×0.5° (longitude, latitude) resolution, and over the 17 vertical levels of CHIMERE.**

**Considering its short lifetime, there is no boundary conditions for $NO_2$. For the $NO_x$ inversion, the control vector x is:**

- **the NO and $NO_2$ anthropogenic emissions at a 1-day temporal resolution, at a 0.5° ×0.5° (longitude, latitude) resolution and over the first 8 vertical levels, i.e. for each of the corresponding 101×85×8 grid cells,**
- **the NO and $NO_2$ 3D initial conditions for the 19$^{th}$ February 2015 at 0:00 UTC, at a 0.5° ×0.5° (longitude, latitude) resolution and over the 17 vertical levels of CHIMERE."**

Am I to assume that NOx emissions are estimated only for 1 day, and all days are the same?

**$NO_x$ emissions are indeed estimated only for 1 day in our illustration**, **as now indicated in the beginning of Section 4: "**We have chosen to present an illustration of CO inversion **over seven days,** the first week of March 2015. Considering the short lifetime of $NO_x$ of a few hours [Valin et al., 2013; Liu et al., 2016], we have chosen to present illustration of $NO_x$ inversion **over one day,** 19 February 2015**."**

**All days could use the same inversion set-up with relevant prior emissions and observations.**

For CO, what do you mean by 7-day? Average?

**It is not an average, the same increments are applied for the 7-day window, as seen in the definition of the control vector.**

How are emissions incorporated in CHIMERE in terms of time? Is there a distribution? i.e., diurnal and weekly cycle?

**The anthropogenic emissions are constant within an hour, and the biogenic ones are linearly interpolated within an hour. The temporal distributions of the emissions are already described in the Section 4.1.2 in manuscriptgmd-2019-186-manuscript-version7.pdf: "Temporal profiles are also provided per Gridded Nomenclature For Reporting (GNFR) sector code (variations due to the month, weekday and hour)."**

These points seem unchanged, has the explanation been lost? The table may be not necessary, but some questions are unanswered.

**We have answered all the questions of the point 49. Table 1 is still present in the last version of the manuscriptgmd-2019-186-manuscript-version7.pdf. We have improved the definition of our control vector x, to further help the understanding of our case illustrations.**

51) Section 4.2.3. Is it possible to break down the components of J?
**in gmd-2019-186-author_response-version2.pdf p45:** *The control vector and observation vectors gather fluxes and concentrations (respectively) over the whole data assimilation temporal window. Decomposing the cost function into a sum of comparisons between the projection of the control variables in the observation space and the observations at different times would actually make it quite confusing and uselessly complicated.* **Therefore we prefer not doing it.**

How about emission error reduction?
**in gmd-2019-186-author_response-version2.pdf p46:** *As already discussed in Section 2, posterior error covariances are not a straight forward product of variational inversion systems. We have added a reference to Rayner et al.[2019] in this paragraph.*

**in gmd-2019-186-author_response-version2.pdf p57:** *The emission error reduction is not a straightforward product of the method.***"**

How do you ensure that these increments are "resolved by the observations".
**in gmd-2019-186-author_response-version2.pdf p57:** *The method ensures it by design. Sensitivity tests will be done later over longer periods for example with constant emissions to quantify the impact of the observations in the system.*

It would be great to see error reduction plots, if posterior error covariances are calculated.
**in gmd-2019-186-author_response-version2.pdf p58:** *They aren't, as explained before.*

How about initial conditions? Did this change as well?
**in gmd-2019-186-author_response-version2.pdf p58:***The initial conditions are slightly changed. This is now described in Section 4.2.2: "With prior error standard deviations assigned to 15% of the initial conditions, the changes in initial conditions are very small (not shown) and do not affect the posterior emissions (test B, Figure 8)."*

Only part of the questions are addressed in the text.
**We have answered all the questions of the point 51 and we think that only 2 (about the emission error reduction and about the changes in initial conditions) should lead to changes in the text.**

52) Line 508-516. What is the implication of this to overall cost and computing and optimality of minimization including error correlation of CO and NOX (and spatial correlation against superobbing) as well as increase in dimension of control vector? This also entails using this system at higher spatiotemporal resolution, right? It would be great to have a section on limitations before future implications

Part of the question is still not addressed.

**We have answered above about the error correlation of CO and NO$_x$. We have also answered about TROPOMI high spatial resolution: "***The high-resolution imaging of TROPOMI will indeed entail using PYVAR-CHIMERE at higher spatio-temporal resolutions, but for smaller domains (i.e.,*

*over countries rather than over Europe as a compromise between resolution and the computational cost.***"**

**We agree to insert this sentence in the conclusion of our manuscript: "***These new space missions with high-resolution imaging have the ambition to monitor atmospheric chemical composition for the quantification of anthropogenic emissions*. It will indeed entail using PYVAR-CHIMERE at higher spatio-temporal resolutions, but probably for smaller domains (i.e., over countries rather than over Europe) as a compromise between resolution and the computational cost**."